# ConfLab: A Data Collection Concept, Dataset, and Benchmark for Machine Analysis of Free-Standing Social Interactions in the Wild

**Chirag Raman**[1*]     **Jose Vargas-Quiros**[1*]     **Stephanie Tan**[1*]     **Ashraful Islam**[2]

**Ekin Gedik**[1]     **Hayley Hung**[1]

[1]Delft University of Technology, Delft, The Netherlands
{c.a.raman, j.d.vargasquiros, s.tan-1, e.gedik, h.hung}@tudelft.nl
[2]Rensselaer Polytechnic Institute, New York, USA
islama6@rpi.edu

## Abstract

Recording the dynamics of unscripted human interactions in the wild is challenging due to the delicate trade-offs between several factors: participant privacy, ecological validity, data fidelity, and logistical overheads. To address these, following a *datasets for the community by the community* ethos, we propose the Conference Living Lab (ConfLab): a new concept for multimodal multisensor data collection of in-the-wild free-standing social conversations. For the first instantiation of ConfLab described here, we organized a real-life professional networking event at a major international conference. Involving $48$ conference attendees, the dataset captures a diverse mix of status, acquaintance, and networking motivations. Our capture setup improves upon the data fidelity of prior in-the-wild datasets while retaining privacy sensitivity: 8 videos ($1920 \times 1080$, 60 fps) from a non-invasive overhead view, and custom wearable sensors with onboard recording of body motion (full 9-axis IMU), privacy-preserving low-frequency audio (1250 Hz), and Bluetooth-based proximity. Additionally, we developed custom solutions for distributed hardware synchronization at acquisition, and time-efficient continuous annotation of body keypoints and actions at high sampling rates. Our benchmarks showcase some of the open research tasks related to in-the-wild privacy-preserving social data analysis: keypoints detection from overhead camera views, skeleton-based no-audio speaker detection, and F-formation detection.

## 1   Introduction

A crucial challenge towards developing artificial socially intelligent systems is understanding how *real-life* situational contexts affect social human behavior [1]. Social-science findings indeed show that the dynamics of how we conduct daily interactions vary significantly depending on the social situation [2–4]. Unfortunately, such dynamics are not adequately captured by many data collection setups where role-played or scripted scenarios are typical [5].

In this paper we address the problem of collecting a privacy-sensitive dataset of unscripted social dynamics of real-life relationships where encounters can influence someone's daily life. We argue that doing so requires recording these exchanges in the natural ecology, requiring an approach

---

*Equal contribution

36th Conference on Neural Information Processing Systems (NeurIPS 2022) Track on Datasets and Benchmarks.

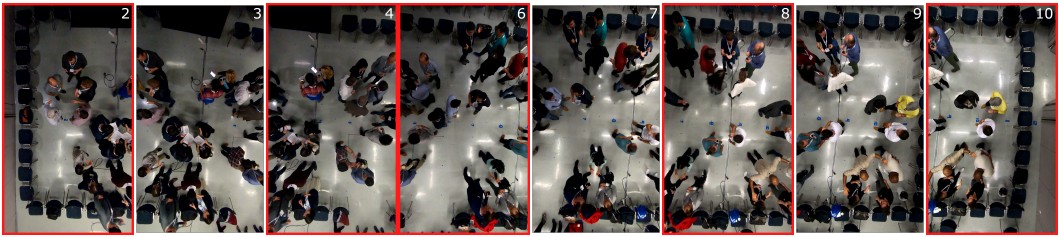

Figure 1: Snapshot of the interaction area from our cameras. We annotated only cameras highlighted with red borders (high scene overlap). For a clearer visual impression of the scene, we omit cameras 1 (few people recorded) and 5 (failed early in the event). Faces blurred to preserve privacy.

different from the typical setup of locally-organized studies. Specifically, we focus on free-standing interactions within the setting of an international conference (see Figure 1).

Recording an international community in its natural habitat is characterized by several intersecting challenges: an intrinsic trade-off exists between data fidelity, ecological validity, and privacy preservation. For ecological validity, a non-invasive capture setup is essential for mitigating any influence on behavior naturalness [6–8].The most common solution involves mounting cameras from aerial perspectives such as top-down [9, 10] and elevated-side views [11–13]. Now elevated-side views make it easy to capture sensitive personal information such as faces, which leads to several ethical concerns. For instance, capturing faces has been related to harmful downstream surveillance applications [14]. Besides, state-of-the-art (SOTA) body-keypoint estimation techniques perform poorly on aerial perspectives [9, 15], making the extraction of automatic pose annotations challenging (Figure 3). To avoid such issues, some researchers have turned to more privacy-preserving wearable sensors shown to benefit many behavior analysis tasks [8, 16, 17].

In all, the closest related datasets (see Table 1) suffer from several technical limitations precluding the analysis and modeling of fine-grained social behavior: (i) lack of articulated pose annotations; (ii) a limited number of people in the scene, preventing complex interactions such as group splitting/merging behaviors, and (iii) an inadequate data sampling-rate and synchronization-latency to study time-sensitive social phenomena [18, Sec. 3.3]. To address all these limitations, we propose the Conference Living Lab (ConfLab): a new concept for multimodal multisensor data collection of ecologically-valid social settings. From the first instantiation of ConfLab, we provide a high-fidelity dataset of 48 participants at a professional networking event.

**Methodological Contributions:** We describe a data collection design that captures a diverse mix of real levels of seniority, acquaintance, affiliation, and motivation to network (see Figure 2). This was achieved by organizing ConfLab as part of a major international scientific conference. ConfLab had these goals: (i) a data collection effort follwing a *by the community for the community* ethos: the more volunteers, the more data, (ii) volunteers who potentially use the data can experience first-hand potential privacy and ethical considerations related to sharing their own data, (iii) in light of recent data sourcing issues [14, 20], we incorporated privacy and invasiveness considerations directly into the decision-making process regarding sensor type, positioning, and sample-rates.

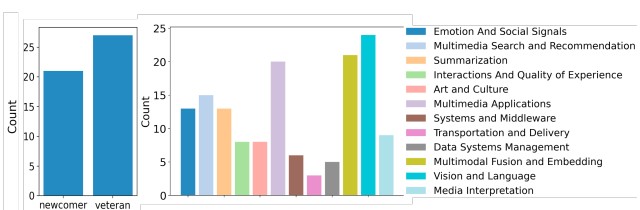

Figure 2: Frequency of newcomer/veteran participants (left) and reported research interests (right).

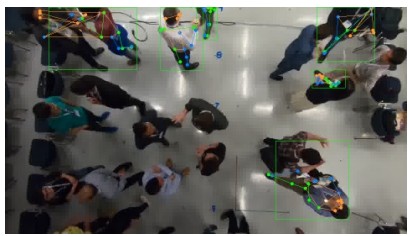

Figure 3: Keypoint detection using pretrained RSN [19]. Additional SOTA results are in Appendix F.1

Table 1: Comparison of ConfLab with prior datasets of free-standing conversation groups in-the-wild social interaction settings. Conflab is the first and only social interaction dataset that offers skeletal keypoints and speaking status at high annotation resolution, as well as hardware synchronized camera and multimodal wearable signals at high resolution.

| Dataset | People/ Scene | Video | Manual Annotations | Wearable Signals | Synchronization |
|---|---|---|---|---|---|
| Cocktail [13]† | 7 | $512 \times 384$ | F-formations (20 and 30 min, 1/5 Hz) | None | Unknown |
| CoffeeBreak [12] | 14 | $1440 \times 1080$ | F-formations (130 frames in two sequences) | None | None |
| IDIAP [10] | > 50 | 180 min; $654 \times 439$ 20 fps | F-formations (82 independent frames) | None | None |
| SALSA [11]† | 18 | 60 min; $1024 \times 768$ 15 fps | Bounding boxes (30 min) Head & body ori. (30 min) F-formations (60 min) (all 1/3 Hz) | Audio MFCCs (30 Hz) Acceleration (20 Hz) IR proximity (1 Hz) | Post-hoc infra-red event-based (no-drift assumption) |
| MnM [9]† | 32 | 30 min; $1920 \times 1080$ 30 fps | Bounding boxes (30 min, 1 Hz ‡ ) F-formations (10 min, 1 Hz ) Actions (45 min, 1 Hz‡) | Accelerometer (20 Hz) Radio proximity (1 Hz) | Intra-wearable sync via gossiping protocol; Inter-modal sync using manual inspection @1 Hz |
| ConfLab | 48 | $\sim$ 45 min; $1920 \times 1080$ **60 fps** | **17 keypoints (16 min, 60 Hz)** F-formations (16 min, 1 Hz) **Speaking status (16 min, 60 Hz)** | **Low-freq. audio (1250 Hz)** **BT proximity (5 Hz)** **9-axis IMU (56 Hz)** | **Wireless hardware sync at acquisition, max latency of $\sim$ 13 ms** [18] |

† Includes self-assessed personality ratings     ‡ Upsampled to 20 Hz using Vatic [25]     BT: Bluetooth     IMU: Inertial Measurement Unit

**Technical Contributions:** **(i) aerial-view articulated pose**: our annotations of 17 full-body keypoints enable improvements in (a) pose estimation and tracking, (b) pose-based recognition of social actions (under-explored in the top-down perspective), (c) pose-based F-formation estimation (has not been possible from prior work [10, 21–23]), and (d) the direct study of interaction dynamics using full body poses (previously limited to lab settings [24]). **(ii) subtle body dynamics**: we are the first to use a full 9-axis Inertial Measurement Unit (IMU) enabling a richer representation of behaviour at higher sample rates; previous rates were found to be insufficient for downstream tasks [17]. **(iii) enabling finer temporal-scale research questions**: a sub-second crossmodal latency of $\sim$ 13 ms along with higher sampling rate of features (60 fps video, 56 Hz IMU) opens the gateway for the in-the-wild study of nuanced time-sensitive social behaviors like mimicry and synchrony.

## 2 Related Work

Early datasets of in-the-wild social events either spanned only a few minutes (e.g. Coffee Break [12]), or were recorded at such a large distance from the participants that performing robust, automated person detection or tracking with SOTA approaches was non-trivial (e.g. Idiap Poster Data [10]). More recently, two different strategies have emerged to circumvent such issues.

One approach involves fully instrumented labs with a high resolution multi-camera setup for video and audio data. Here automatic detectors [24, 26, 27] could be applied to obtain poses. This circumvents the cost- and labor-intensive process of manually labeling head poses, at the cost of less portable sensing setups. Notable examples of such in-the-lab studies include seated scenarios, such as the AMI meeting corpus [28], and more recently standing scenarios like the Panoptic Dataset [24]. Both enable the learning of multimodal behavioral dynamics. However, the dynamics of seated, scripted, or role-playing scenarios are different from that of an unconstrained social setting such as ours. In contrast, ConfLab moves out of the lab with a more modular and portable multimodal, multisensor solution that scales easily in the wild.

Another approach exploited wearable sensor data to allow for multimodal processing—sensors included 3 or 6 DOF inertial measurement units (IMU); infrared, bluetooth, or radio sensors to measure proximity; or microphones for speech behavior [9, 11]. While proximity has been used as a proxy of face-to-face interaction [11, 29–32], recent findings highlight significant problems with such an assumption [33]. Such errors can have a significant impact on the machine-perceived experience of an individual, precluding the development of personalized technology. Chalcedony badges used by

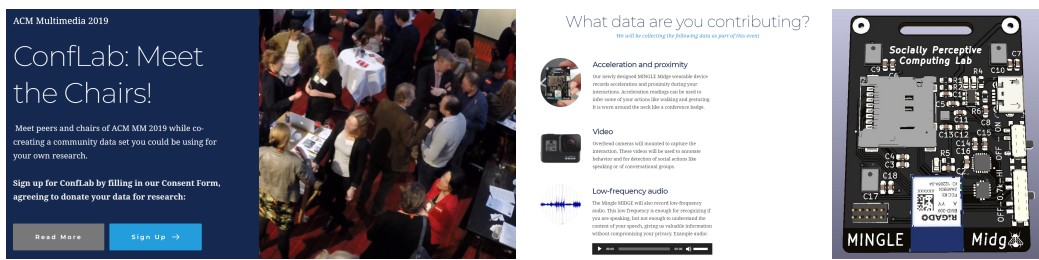

Figure 4: Screenshots from the *ConfLab: Meet the Chairs!* event website    Figure 5: The Midge

[9] show more promising results with a radio-based proximity sensor and accelerometer [34], but such data remains insufficient for more downstream tasks due to the relatively low sample (20Hz) and annotation (1Hz) frequency [17]. In light of these challenges in wearable sensing, ConfLab features custom-developed Midge sensors that enable more flexible and fine-grained on-device recording. At the same time, ConfLab enables researchers in the wearable and ubiquitous computing communities to investigate the benefit of exploiting wearable and multimodal data.

Furthermore, while both SALSA [11] and MatchNMingle [9] capture a multimodal dataset of a large group of individuals involved in mingling behavior, the inter-modal synchronization is only guaranteed at 1/3 Hz and 1 Hz, respectively. Prior works coped with lower tolerances by computing summary statistics over input windows [17, 35, 36]. While 1 Hz is able to capture some conversation dynamics [37], it is insufficient to study fine-grained social phenomena such as back-channeling or mimicry that involve far lower latencies [18, Sec. 3.3]. ConfLab provides data streams with higher sampling rates, synchronized at acquisition with our method shown to yield a 13 ms latency at worst [18] (see Sec. 3). Table 1 summarizes the differences between ConfLab and other related datasets.

## 3   Data Acquisition

In this section we describe the considerations, design, and supporting community engagement activities for the first instantiation of ConfLab at ACM Multimedia 2019 (MM'19), to serve as a template and case study for other similar efforts.

**Ecological Validity and Recruitment**   An often-overlooked but crucial aspect of in-the-wild data collection is the design and ecological validity of the interaction setting [6–8]. To capture natural interactions in a professional setting and encourage mixed levels of status, acquaintance, and motivations to network, we co-designed a networking event with the MM'19 organizers called *Meet the Chairs!* Our event website (https://conflab.ewi.tudelft.nl/) served to inform participants about the goals of a community created dataset, and transparently describe the data collection process (Figure 4). During the conference, participants were recruited via word-of-mouth marketing, social media, conference announcements, and the event website. As an additional incentive beyond interacting with the Chairs and participating in a community-driven data endeavor, we provided attendees with post-hoc insights into their networking behavior from the collected wearable-sensors data. See Supplementary material for a sample participant report.

**Privacy and Ethics**   The collection and sharing of ConfLab is GDPR compliant. The dataset design and process was approved by both, the Human Research Ethics Committee (HREC) at our institution (TUDelft) and the conference location's national authorities (France). All participants gave consent for the recording and sharing of their data at registration.(See the Datasheet in the Appendix for the consent form.) Given the involvement of private human data, ConfLab is only available for academic research purposes under an End User License Agreement. Such an *as open as possible and as closed as necessary* ethos for open science acknowledges the limitation that personal data places on open sharing [38, 39].

**Data Capture Setup**   Our goal while designing the capture setup was to find the best trade-off between maximizing data fidelity and interfering with the naturalness of the interaction (ecological validity) or violating participant privacy (ethical considerations). Through discussions with the HREC and General Chairs of MM'19 we decided to mitigate the capture of faces, which constitute one of the

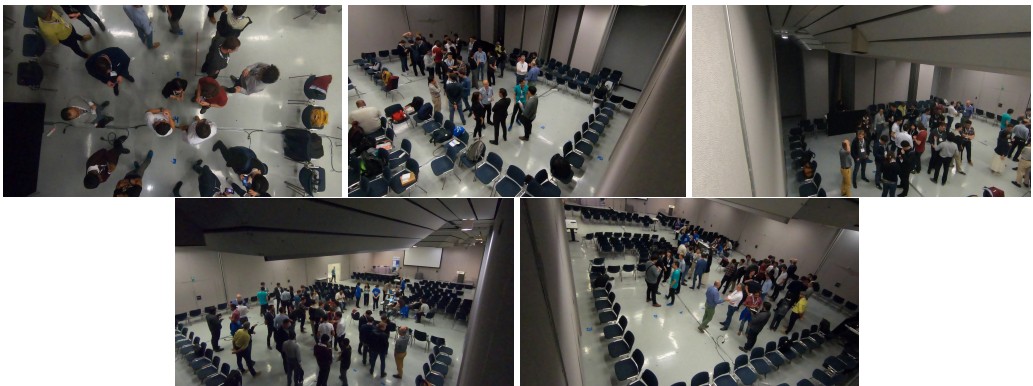

Figure 6: Comparing the top-down (top-left, camera 4) and elevated-side camera views (rest). Note how the top-down view is better at mitigating the capture of faces and suffers from fewer occlusions. This allows for a clearer capture of gestures and lower extremities for the most number of people while also preserving privacy.

most sensitive personally-identifiable features. Avoiding the inclusion of faces serves two purposes. First, it safeguards against misuse in downstream tasks with potential negative societal impacts such as harmful surveillance. Such issues have led to the retraction of some person re-identification datasets [14]. Second, it protects the participants who are part of a real research community; since the dataset does not involve role-playing or scripted conversations, the dataset contains their actual behavior. Consequently, we chose an aerial perspective for the video modality (see Figure 6). The 10 m × 5 m interaction area was recorded by 14 GoPro Hero 7 Black cameras (60fps, 1080p, Linear, NTSC) [40]. 10 of these were placed directly overhead at a height of ∼ 3.5 m at 1 m intervals, with 4 cameras at the corners providing an elevated-side-view perspective. (The HREC has suggested not sharing the elevated-side-view videos due to the presence of faces.) For capturing multimodal data streams, we designed a custom wearable multi-sensor pack called the Midge[2] (see Figure 5 for a design render), based on the open-source Rhythm Badge designed for office environments [41]. We improved upon the Rhythm Badge to achieve more fine-grained and flexible data capture (see Appendix D). We designed the Midge in a conference badge form-factor for seamless integration. Unlike smartphones, wearable badges allow for a simple *grab-and-go* setup and do not suffer from sensor/firmware differences across models. Popular human behavior datasets are synchronized by maximizing similarity scores around manually identified common events, such as infrared camera detections [11], or speech plosives [42]. While recordings in lab settings can allow for fully wired recording setups, recording in-the-wild requires a distributed wireless solution. We developed a solution to synchronize the cameras and wearable sensors directly at acquisition while significantly lowering the cost of the recording setup [18], making it easier for others to replicate our capture setup. See Appendix D for synchronization and calibration details, and Appendix B for images of the setup.

**Data Association and Participant Protocol**   One consideration for multimodal data recording is the data association problem—how can pixels corresponding to an individual be linked to their other data streams? To this end, we designed a participant registration protocol. Arriving participants were greeted and fitted with a Midge. The ID of the Midge acted as the participant's identifier. One team member took a picture of the participant while ensuring both the face of the participant and the ID on the Midge were visible. In practice, it is preferable to avoid this step by using a fully automated multimodal association approach. However this remains an open research challenge [43, 44]. During the event, participants mingled freely—they were allowed to carry bags or use mobile phones. Conference volunteers helped to fetch drinks for participants. Participants could leave before the end of the one hour session.

**Replicating Data Collection Setup and Community Engagement**   After the event, we gave a tutorial at MM'19 [45] to demonstrate how our collection setup could be replicated, and to invite conference attendees and event participants to reflect on the broader considerations surrounding privacy-preserving data capture, sharing, and future directions such initiatives could take.

---

[2]Documentation and schematics: https://github.com/TUDelft-SPC-Lab/spcl_midge_hardware

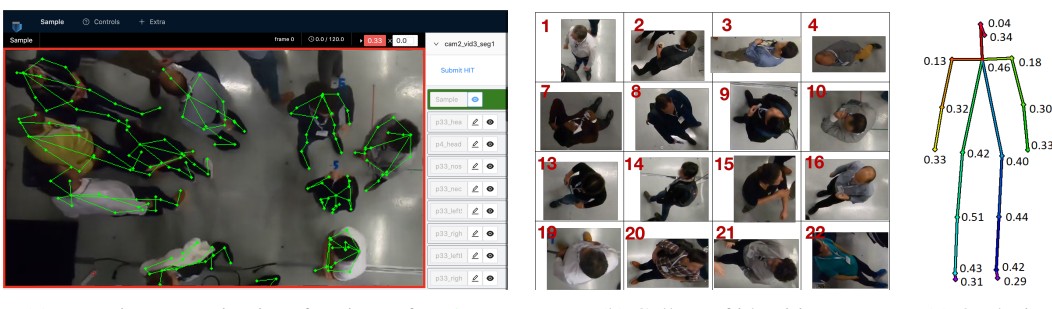

(a) Keypoint annotation interface in covfee [47]        (b) Gallery of identities        (c) Occlusion

Figure 7: Illustration of the body keypoints annotation procedure: (a): our custom time continuous annotation interface; (b): the gallery of person identities used by annotators to identify people in the scene (faces blurred); and (c): the skeleton template with the fraction of occluded frames.

# 4 Data Annotation

**Continuous Keypoints Annotation**    Existing datasets of in-the-wild social interactions have mainly focused on localizing subjects via bounding boxes [9, 11]. However, richer information about the social dynamics such as gestures and changes in orientation cannot be retrieved from bounding boxes alone, and necessitates the labeling of multiple skeletal keypoints. The typical approach to keypoint annotation involves using tools such as Vatic [25] or CVAT [46] to manually label every $N$ frames followed by interpolating over the rest of the frames. This one-frame-at-a-time annotation procedure makes obtaining keypoint annotations a labor- and cost-intensive process. Moreover, interpolation fails to capture the finer temporal dynamics of the underlying behavior, and reduces the benefits of higher-framerate video capture. Limited by existing tools, no related dataset of in-the-wild human behavior has included time-continuous pose or speaking status annotations.

In contrast, to overcome these issues we collected fine-grained time-continuous annotations of keypoints via a web-based interface implemented as part of the Covfee framework [48]. Here, annotators follow individual joints using their mouse or trackpad while playing the video in their web browser. The playback speed of the video is automatically adjusted using an optical-flow-based technique to enable annotators to follow keypoints continuously without pausing the video. This design enables easy keypoint labeling in *every* frame of the video (60 Hz). We also incorporated a binary *occlusion* flag for every body keypoint. Annotators simultaneously controlled this flag to indicate when a body joint was not directly visible. Note that the flag is only an additional confidence indicator; we asked the annotators to label the occluded keypoint using their best estimate if it was deemed to be within the frame. Our pilot study on the efficacy of Covfee compared to non-continuous annotation via CVAT [46] is presented in [48]. For the pilot annotators, the continuous annotation methodology resulted in a $3\times$ speedup with statistically indifferent error rates.

We chose the top-down camera views for annotation since they suffer from fewer occlusions than the elevated-side views, enabling improved capture of gestures and lower extremities for more number of people (see Figure 6). Given the overlap in the camera views, we annotated keypoints in five of the ten overhead cameras (see Figure 1). Note that the same subject could be annotated in multiple cameras due to the overlap in even the five annotated cameras. Videos were split into two-minute segments to ease the annotation procedure. Each segment was annotated by one annotator by tracking the joints of all the people in the scene.

**Continuous Speaking Status Annotations**    Speaking status is a key non-verbal cue for many social interaction analysis tasks [49]. We annotated the binary speaking status of every subject due to its importance as a key feature of social interaction [16, 50–53] and to contribute the existing community who are working on this task [17, 54, 55]. Action annotations have traditionally been carried out using frame-wise techniques [9], where annotators find the start and end frame of the action of interest using a graphical interface. Given the speed enhancement of continuous annotation, we also annotated speaking status via a continuous technique. We implemented a binary annotation interface as part of Covfee [48]. We asked annotators to press a key when they perceived speaking starting or ending. In a pilot study with two annotators, we measured a frame-level agreement (Fleiss' $\kappa$) of

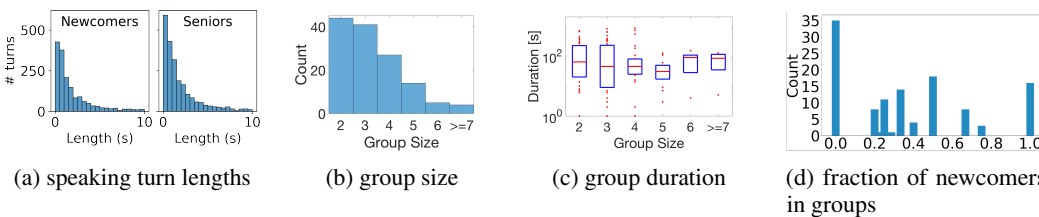

(a) speaking turn lengths  (b) group size  (c) group duration  (d) fraction of newcomers in groups

Figure 8: Data distributions for speaking status and conversation groups

0.552, comparable to previous work [35]. Similar to [9], the annotations were made by watching the video. We provided the annotators with all overhead views to best capture visual behavior.

**F-formation Annotations**   Identifying who is likely to have social influence on whom is another important feature for analyzing social behavior. This is operationalised via the theory of F-formations, which are groups of people arranging themselves to converse or socially interact. Similar to prior datasets [9, 11, 13], F-formations group membership were annotated using an approximation of Kendon's definition [56]. F-formation stands for Facing formation, which is a socio-spatial arrangement where people have direct, easy and equal access while excluding the space from others in the surroundings. The arrangement commonly maintains a convex space in the middle of all the participants (determined by the location and orientation of their lower body), although other spatial arrangements (e.g., side-by-side, L-shaped) are possible, especially for smaller-sized groups of people. Annotations were labeled by one annotator at 1 Hz, following this definition. Since this is a largely objective and common framework for defining F-formations, we deemed it sufficient to obtain one set of annotations. Further, since F-formations may span camera views, we always used the camera that captured each F-formation in its entirety for annotation.

## 5   Dataset Statistics

**Individual-Level Statistics**   Figure 7c shows the average occlusion values we obtained from annotators for each of the 17 keypoints. In Figure 8a we show the distribution of turn lengths in our speaking status annotations, for both newcomers and veterans, as per their self-reported newcomer status to the conference. We defined a turn to be a contiguous segment of positively-labeled speaking status, which resulted in a total of 4096 turns annotated.

**Group-Level Statistics**   We found 119 distinct F-formations of size greater than or equal to two, and 38 instances of singletons. Of these, there are 14 F-formations and 2 singletons that include member(s) using the mobile phone. The distributions for group size and duration per group size are shown in Figure 8b and Figure 8c, respectively. Mean group duration doesn't seem to be influenced by group size although higher variations are seen at smaller group sizes. The fraction of community newcomers (first-time attending the conference) in groups is summarized in histogram in Figure 8d. The figure demonstrates two peaks on both sides of the spectrum (i.e., no newcomers vs. all newcomers in the same group). This spread over mixed and non-mixed seniority presents opportunities to study how acquaintance and seniority influence conversation dynamics.

## 6   Research Tasks

We report experimental results on three baseline benchmark tasks: person and keypoints detection, speaking status detection, and F-formation detection. The first task is a fundamental building block for automatically analyzing human social behaviors. The other two demonstrate how learned body keypoints can be used in the behavior analysis pipeline. We chose these benchmarking tasks since they have been commonly studied on other in-the-wild behavior datasets. Code for all benchmark tasks is available at: `https://github.com/TUDelft-SPC-Lab/conflab`. See the *Uses* section of the Datasheet in the Appendix for a discussion of the broader range of tasks ConfLab enables.

Table 2: Mask-RCNN results for person bounding box detection and keypoint estimation.

| Model | Person Detection | | | Keypoint Estimation | | |
|---|---|---|---|---|---|---|
| | $AP_{50}$ | AP | $AP_{75}$ | $AP_{50}^{OKS}$ | $AP^{OKS}$ | $AP_{75}^{OKS}$ |
| R50-FPN | 73.9 | 38.9 | 38.4 | 45.3 | 13.5 | 3.3 |

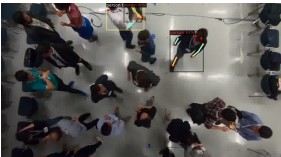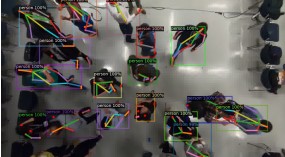

Figure 9: Predictions from the Mask-RCNN model; COCO pretrained (left), and ConfLab finetuned (right).

## 6.1 Person and Keypoints Detection

This benchmark involves the tasks of person detection (identifying bounding boxes) and pose estimation (localizing skeletal keypoints). Since pre-trained SOTA methods struggle with a privacy-sensitive top-down perspective [15] (also see Figure 3 and Appendix F.1 for ConfLab results), we finetune COCO-pretrained models on our dataset. We used Mask-RCNN [57] (Detectron2 framework [58] implementation) with a ResNet-50 backbone for both tasks for benchmarking. Since keypoint annotations were made per camera, we used four of the overhead cameras for training (Cameras 2, 4, 8, 10) and one for testing (Camera 6). Implementation details are available in Appendix E.1.

**Evaluation Metrics** We evaluated person-detection performance using the standard metrics in the MS-COCO dataset paper [59]. We report average precision (AP) for intersection over union (IoU) thresholds of 0.50 and 0.75, and the mean AP from an IoU range from 0.50 to 0.95 in 0.05 increments. For keypoint detection, we use object keypoint similarity (OKS) [59]. $AP^{OKS}$ is a mean average precision for different OKS thresholds from 0.5 to 0.95.

**Results and Analyses** Table 2 summarizes our person detection and joint estimation results. Our baseline achieves 73.9 $AP_{50}$ in detection and 45.3 $AP_{50}^{OKS}$ in keypoint estimation. Figure 9 shows qualitative results from our fine-tuned network. For further insight we performed several analyses and ablations. In Appendix Table 6, we depict the effect of varying the number of training samples on performance. For training, we use the same four cameras and only vary the number of frames for each camera. We evaluate on the same testing images from camera 6. We find that performance saturates at 16% training samples. We next investigated the effect of increasing training data size by adding specific cameras one at a time. We report results in Appendix Table 7. There is a 260% performance gain when first doubling the training samples to 69 k with the addition of camera 4, and a 46% gain when adding another 43 k samples from camera 8. Finally, since the lower body regions suffer from higher occlusion, we experiment with different sections of body for further insight and report results in Appendix Table 8.

## 6.2 Speaking Status Detection

In data collected from real-life social settings, individual audio recordings can be hard to obtain due to privacy concerns [60]. This has led to the exploration of other modalities to capture some of the motion characteristics of speaking-related gestures [35, 36]. In this task we explore the use of body pose and wearable acceleration data for detecting the speaking status of a person in the scene.

**Setup** We use the SOTA MS-G3D graph neural network for skeleton action recognition [61], pre-trained on Kinetics Skeleton 400. For the acceleration modality, we evaluated three time series classifiers, each of which we trained from scratch: 1D Resnet [62], InceptionTime [63], and Minirocket [64]. We performed late fusion by averaging the scores from both modalities. Like prior work [17, 36], the task was set up as a binary classification problem. We divided our pose (skeleton) tracks into 3-second windows with 1.5 s overlap. A window was labeled positive if more than 50% of the continuous speaking status labels within it are positive. This resulted in an imbalanced dataset of 42882 windows with 29.2% positive labels. Poses were pre-processed for training following [61]. Three of the keypoints (head, and feet tips) were discarded due to not being present in Kinetics. We adapted the network by freezing all layers except for the last fully connected layer and training for five extra epochs. Acceleration readings were not pre-processed, other than by interpolating the original variable-sampling-rate signals to a fixed 50 Hz.

Table 3: ROC AUC and accuracy of skeleton-based, acceleration-based and multimodal speaking status detection (10-fold cross-validation).

| Modality | Model | AUC | Acc. |
|---|---|---|---|
| Pose | MS-G3D [66] | 0.676 | 0.677 |
| | InceptionTime [63] | 0.798 | 0.768 |
| Acceleration | Resnet 1D [62] | 0.801 | 0.767 |
| | Minirocket [64] | 0.813 | 0.768 |
| Multimodal | MS-G3D + Minirocket | 0.823 | 0.775 |

Table 4: Average F1 scores for F-formation detection comparing GTCG [23] and GCFF [67] with the effect of different threshold and orientations (standard deviation in parenthesis).

| | GTCG | | GCFF | |
|---|---|---|---|---|
| | T=2/3 | T=1 | T=2/3 | T=1 |
| Head | 0.51 (0.09) | 0.40 (0.12) | 0.47 (0.07) | 0.31 (0.23) |
| Shoulder | 0.46 (0.11) | 0.38 (0.11) | 0.56 (0.25) | 0.36 (0.16) |
| Hip | 0.45 (0.10) | 0.37 (0.12) | 0.39 (0.06) | 0.25 (0.11) |

**Evaluation**  Evaluation was carried out via 10-fold cross-validation at the subject level, ensuring that no examples from the test subjects were used in training. We used the area under the ROC curve (AUC) as main evaluation metric to account for the imbalance in the labels.

**Results**  The results in Table 3 indicate a better performance from the acceleration-based methods. One possible reason for the lower performance of the pose-based methods is the significant domain shift between Kinetics and Conflab, especially in camera viewpoint (frontal vs top-down). The acceleration performance is in line with previous work [17]. Multimodal results were slightly higher than acceleration-only results, despite our naive fusion approach, a possible point to improve in future work [65]. Experiments with the rest of the IMU modalities are presented in Appendix F.2.

### 6.3 F-formation Detection

**Setup**  Like prior work [10, 21–23], we operationalize interaction groups using the framework of F-formations [56]. We provide performance results for F-formation detection using GTCG [23] and GCFF [67] as a baseline. Recent deep learning methods such as DANTE [22] are not directly applicable since they depend on knowing the number of people in the scene, which is variable for ConfLab. We use pre-trained model parameters (reported in the original GTCG and GCFF papers on the Cocktail Party dataset [13]) and tuned a subset of parameters more relevant to ConfLab attributes on camera 6. More details can be found in Appendix E.2. We derive three different sets of orientation features from (i) head, (ii) shoulder and (iii) hip keypoints.

**Evaluation Metrics**  We use the standard F1 score as evaluation metric for group detection [23, 67]. A group is correctly estimated (true positive) if at least $\lceil T * |G| \rceil$ of the members of group $G$ are correctly identified, and no more than $1 - \lceil T * |G| \rceil$ is incorrectly identified, where $T$ is the tolerance threshold. We report results for $T = \frac{2}{3}$ and $T = 1$ (more strict threshold) in Table 4.

**Results**  We show that different results are obtained using different sources of orientations. Different occlusion levels in keypoints due to camera viewpoint may have affected performance. Another factor influencing model performance is that F-formations (which are driven by lower-body orientations [56]) may have multiple conversations floors [51]. Floors are indicated by coordinated speaker turn taking patterns and influence coordinated head orientations of the group.

## 7   Conclusion and Discussion

ConfLab contributes a new concept for real-life data collection in the wild and captures a high-fidelity dataset of mixed levels of acquaintance, seniority, and personal motivations.

**ConfLab: the Dataset**  We improved upon prior work by providing higher-resolution, fidelity, and synchronization across sensor networks. We also carefully designed our social interaction setup to enable a diverse mix of seniority, acquaintanceship, and motivations for mingling. The result is a rich set of 17 body-keypoint annotations of 48 people at 60 Hz from overhead cameras for developing more robust estimation of keypoints, speaking status and F-formations for further analyses of more complex socio-relational phenomena. Our benchmark results for these tasks highlight how the improved fidelity of ConfLab can assist in the development of more robust methods for these key tasks. We hope that models trained on ConfLab for localizing keypoints would fill the gap in the cue

extraction pipeline, enabling past datasets [9, 10] without articulated pose data to be reinvigorated; this would open the floodgates for more robust analysis of the social phenomena labeled in these other datasets. Finally, our baseline social tasks form the basis for further explorations into downstream prediction tasks of socially-related constructs such as conversation quality [68], dominance [53], rapport [50], influence [69] etc.

**ConfLab: the Data-Collection Concept**    To relate an individual's behaviors to trends within their social network, further iterations of ConfLab are needed. These iterations would enable the study of behavioral patterns at different timescales, including multiple interactions in one day, multiple days at a conference, or across distinct conferences. This paper serves as a template for such future ventures. We hope that if the idea of a conference as a living lab gains traction, the effort and cost of data collection can be amortized across different research groups, even involving support from the conference organizers. This *data by the community for the community* ethos can enable the generation of a corpus of related datasets enabling new research questions.

**Societal Impact**    ConfLab's long-term vision is towards developing technology to assist individuals in navigating social interactions. In this work we have identified choices that maximize data fidelity while upholding ethical best practices: an overhead camera perspective that mitigates identifying faces, recording audio at a low-frequency, and using non-intrusive wearable sensors matching a conference badge form-factor. We argue this is an essential step towards a long-term goal of developing personalized and socially aware technologies that enhance social experiences. At the same time, such interventions could also affect a community in unintended ways: worsened social satisfaction, lack of agency, stereotyping; or benefit only those members of the community who make use of resulting applications at the expense of the rest. More nefarious uses involve exploiting the data for developing methods that harmfully surveil or profile people. Researchers must consider such inadvertent effects while developing downstream applications. Finally, since we recorded the dataset at a scientific conference and required voluntary participation, there is an implicit selection bias in the population represented in the data. Researchers should be aware that insights resulting from the data may not generalize to the general population.

**Empowering Users Through an Agentist Rather Than Structurist Approach**    The analysis of human behavior in social settings has classically taken a more top-down perspective. For instance, the analysis of situated interactions (via only proximity networks) has provided insight into the process of making science in the field of Meta Science [70]. However, while social network science is a well-populated domain, it lacks a more individualized measurement of social behavior: see more discussion of the structure vs. agency debate [71]. Relying on the network science approach jeopardizes an individual's right to technologies that enable free will. We consider the agency in choosing such technologies to be a form of individual harm avoidance. ConfLab provides access to more than just proximity data about social interactions, enabling the study of context-specific social dynamics. These dynamics are a uniquely dependent not only on the individual, but also the group they are interacting with [72]. We hope our highlighting of participatory design practices and these value-sensitive design principles promote social safety in developing socially assistive technologies.

# Acknowledgements

The authors would like to thank: the ACM Multimedia 2019 General Chairs Martha Larson, Benoit Huet, and Laurent Amsaleg for their support in making the data collection at a major international conference a reality; Bernd Dudzik, Yeshwanth Napolean, Ruud de Jong, and the venue support staff for their help in setting up the recording on site; Ioannis Protonotarios for the development of the MINGLE Midge badge; Jerry de Vos for improving our Midge Github repository and designing a new case; the participants and student volunteers for the *Meet the Chairs!* event; the Amazon Mechanical Turk workers for their efforts in annotating the dataset; Rich Radke, Martin Atzmueller, Laura Cabrera-Quiros, Alan Hanjalic, and Xucong Zhang for the insightful discussions; Santosh Ilamparuthi for the innumerable discussions and support towards strengthening the ethical soundness of recording and sharing ConfLab; Jan van der Heul for the incredibly responsive support in setting up the 4TU Data repository for ConfLab; and Bart Vastenhouw, Myrthe Tielman, and Catharine Oertel for help with the data sharing; and Musy Ayoub for the word-intelligibility analysis of the low frequency audio.

ConfLab was partially funded by Netherlands Organization for Scientific Research (NWO) under project number 639.022.606 with associated Aspasia Grant, and also by the ACM Multimedia 2019 conference via student helpers, and crane hiring for camera mounting.

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
