# OpenReview forum: "ConfLab: A Data Collection Concept, Dataset, and Benchmark for Machine Analysis of Free-Standing Social Interactions in the Wild"
_NeurIPS.cc/2022/Track/Datasets_and_Benchmarks — NeurIPS 2022 Datasets and Benchmarks _

### Official Review · Reviewer_JhBB · 2022-07-19
**Valuable data collection concept, dataset, and benchmark for capturing, annotating, and analyzing free-standing social interactions.**

**Rating:** 8
**Confidence:** 4

**Strengths:**

ConfLab as a data collection concept that allows capturing rich social interaction data in a realistic, yet privacy-preserving setting in semi-controlled venues constitutes a valuable addition to the methodological toolkit of "research on the community by the community" (and human behavior research more generally).

The dataset and benchmark will facilitate progress in computer vision research on (and likely also beyond) the benchmark tasks.

The dataset (and interaction data captured in a ConfLab more generally) could also prove useful for other subcommunities in data mining, machine learning, and human interaction research; for example, the derivable social interaction networks are much richer than most readily available social interaction networks, which often determine interaction only by sensor proximity.

**Weaknesses:**

The overall ConfLab setup appears to be costly (in terms of the financial, temporal, computational, and human resources needed), which limits the replicability of the setup to well-funded institutions or venues.

The work does not discuss potential negative societal impacts in the main paper, but such a discussion (in a spot more prominent than the Appendix) seems warranted, as the work might contribute to the development of privacy invasion and surveillance technologies in a more direct fashion than other datasets and benchmarks.
See also Point 2 in the "Ethics" section.

Also, "in-the-wild" is over-advertising.

**Additional Feedback:**

Content
- Title: "In-the-Wild" seems to be a bit over-advertised (also everywhere else it appears); the setup is more like "In-the-Zoo". You might want to eliminate "In-the-Wild" (if you keep it, it would be "in the Wild" when not used to describe a noun or a verb directly), and perhaps add "Realistic" or "Unscripted" before "Free-Standing" if you feel that this adjective does not convey enough wild(er)ness. However, more generally, the title, which mentions the dataset only, is a bit at odds with your actual contributions: dataset, benchmark, and data collection concept. The latter seems to be most important to you (in particular when judging from the abstract and the conclusion), while the former two might be most relevant for the audience of the NeurIPS Datasets and Benchmarks track – so maybe all three should go into the title? Something in the direction of "ConfLab: A Data Collection Concept, Dataset, and Benchmark for Capturing, Annotating, and Analyzing Free-Standing Social Interactions"?
- l. 21/22: I'm afraid I don't follow the reasoning in this sentence. (1) What are "interactive social systems"? (2) Why is "studying social human behavior in real-life situations" a "crucial challenge" in their "development"? This might be just a problem of phrasing, but your first sentence is currently one of the least clear sentences in the paper (at least to me).
- l. 122: How is the data from the 4 cameras that are not shared to preserve privacy used in the dataset generation process?
- l. 170–173: How do you catch annotation errors or inaccuracies? Can you explain why you deemed it sufficient to have each segment annotated by _one_ annotator only (e.g., why not 2, + 1 for reconciliation)? Was there any kind of quality control? What would be the impact of uncaught inaccuracies in the annotation process?
- Section 5 (related to ConfLab as a Data-Collection Concept): What about participant demographics? You distinguish between newcomers and veterans, but from, e.g., a social science perspective, it would be extremely interesting to see how conversation groups relate to demographic features.
- Figure 6c could become a panel of Figure 7 (it appears a bit out-of-place in Figure 6)
- Figure 2 in the main paper, Figure 11 in the Appendix, Figures in Sample Participant Report: Pie charts are never the answer to any visualization task, unless the task is to visualize the division of a circular pie (or pizza). You are visualizing a 1D statistic in 2D, the size relationship between the pieces now depends on the chosen radius of the pie chart, too, people are bad at judging angles and circularly arranged proportions, ...even worse: the pie chart with an enlarged slice stuck out for emphasis (Sample Participant Report Figures 1, 2, and 5 (a)). I would recommend to use bar charts instead – both for the parts you can still change in the paper and for the participant reports going forward.
- Main paper and Appendix: You seem to be using red, pink, and indian red as colors for emphasis/links. Settle for one, and perhaps make it a shade of blue. Reds and pinks are extremely aggressive to the eye (see also the next point).
- Appendix B: Please change the color of the headers of the datasheet questions and headers from what appears to be "indian red" (#E7485D) to something that is not overwhelming for people with a very sensitive visual cortex. Frankly, there is no reason to use color for emphasis _at all_: The main questions are already bold, the follow-up questions could be typeset in Italics (if you feel the need to distinguish them from your answer by more than the start of a new paragraph), and the block backgrounds could become black or some shade of gray. This is also better for printing (some people might still do that).
- Appendix ll. 801 et seq.: A tabular overview of the costs involved in a ConfLab run (broken down by the steps of the data lifecycle), including an estimate of the total cost, would be appreciated, as this will help others decide whether they can afford it.
- Appendix ll. 832–838: Which platform was used for recruiting crowd workers? What is known about their demographics (esp.: how does your average hourly payment compare to the minimum wage paid in their country of origin)? What was the qualification task?
- Appendix l. 1014: I would recommend using an institutional email address not bound to a person's name here. That way, you can always forward to the currently responsible person without (having to remember) updating the data sheet.

End-User License Agreement (EULA)
You require the signee of the EULA to have a permanent position at their institution, effectively requiring even senior researchers employed as PostDocs to run the form by their formal bosses.
Could you elaborate on the reasoning behind this?

Typos etc. (just those which I found passing by)
- l. 6: "conversations" -> "interactions" (?)
- l. 51: "sample-rates" -> "sampling rates" (?)
- l. 99: comma after "In this section"
- l. 161: comma after "issues"
- l. 177: eliminate "who are" (community is singular)
- l. 200: "the mobile phone" -> "a/their mobile phone" (?)
- l. 229: "For further insight performed" -> "For further insight, we performed"
- l. 244: comma after "task"
- l. 260: "as main evaluation metric" -> "as our main evaluation metric"
- l. 294: excess "e" in "development"
- l. 297: "open the floodgates for" -> "open the door to" (floods have a negative connotation)
- l. 300: excess space after [66]
- l. 300: no need for the "etc" (you already say "such as" in the previous line)
- l. 301: "Data-Collection" -> "Data Collection"
- l. 302: comma after "e.g."
- l. 305: "Regarding maximizing" -> "To maximize"
- l. 310: "long-term goal for" -> "long-term goal of"
- Appendix l. 828: Full stop missing after "(see Appendix C)".
- Appendix l. 1109: space missing after MSPN

**Clarity:**

The paper is structured well and written well.
See "Additional Feedback" for potential micro-level improvements.

**Correctness:**

The data collection concept is sound.
The data underlying the dataset was adequately captured, and it was annotated in an overall reasonable way (but see the related questions in "Additional Feedback").
The tasks selected for the benchmark are natural, interesting, and challenging.
The baselines, training process, chosen evaluation metrics, and overall experimental setup are reasonable.

**Documentation:**

The work introduces a dataset, a benchmark based on that dataset, and a data collection process, all of which are documented well in the paper's Appendix.

**Ethics:**

1. General Ethical Conduct:
The work revolves around sensitive data, but all concerns arising therefrom appear to have been addressed appropriately.

2. Potential Negative Societal Impacts:
The work might contribute to the development of potentially harmful technologies, for example, by enabling the detection of salient conversational features from overhead camera views only, which could be used for surveillance.
The authors discuss some of the related concerns in their datasheet, but not in the main paper.
As the datasheet contains plenty of boilerplate (which is a problem of the Datasheets for Datasets framework, and not the authors' fault), and as it is unclear how many potential users will read only the paper but not the datasheet (perhaps also due to its framework-induced verbosity), I would appreciate a discussion of the most salient potential harms in the main paper.

**Relation To Prior Work:**

The "Related Work" section situates ConfLab in the relevant literature and highlights what distinguishes the work from previous approaches.

**Summary And Contributions:**

The paper makes three contributions.
(1) It introduces ConfLab as a concept to collect multimodal free-standing social interaction data using multiple sensors in semi-controlled settings, such as organized social gatherings.
(2) It introduces a dataset, gathered during the first ConfLab run at an international conference.
(3) It introduces a benchmark, derived from (2), for the tasks of (a) person and keypoints detection, (b) speaking status detection, and (c) F-formation detection.

---

> ### Author Response · Authors · 2022-08-16
> **Addressing Additional Feedback: Part 2**
>
> > What about participant demographics?
>
> We agree it would be interesting, and had discussed collecting demographics with the General Chairs. The conclusion was to not collect demographics to protect conference attendees from potentially harmful downstream research that could contribute to negative stereotypes. This follows similar reasoning to the one mentioned in our Ethical Considerations point in our common clarification here: [https://openreview.net/forum?id=CNJQKM5cV2o&noteId=kxbOf1oGXu](https://openreview.net/forum?id=CNJQKM5cV2o&noteId=kxbOf1oGXu)
>
> > I would recommend to use bar charts instead – both for the parts you can still change in the paper and for the participant reports going forward.
>
> We agree, thank you for this comment. We’ve converted the pie charts in the rebuttal revision to bar charts. For historical accuracy we’ve left the participant report as is in the Appendix, but will also fix it for future ConfLab endeavours.
>
> > Please change the color of the headers of the datasheet questions and headers.
>
> Apologies for the unpleasant experience, we’ve fixed this in the rebuttal revision: black headers, prefacing questions with “Q.” for separation, and consistent use of cyan for both the links and urls.
>
> > Appendix ll. 801 et seq.: A tabular overview of the costs involved in a ConfLab run (broken down by the steps of the data lifecycle), including an estimate of the total cost, would be appreciated, as this will help others decide whether they can afford it.
>
> Thank you for the suggestion, we've added Table 5 in the Datasheet with the itemized costs and updated the discussion.
>
> > Appendix ll. 832–838: Which platform was used for recruiting crowd workers? What is known about their demographics (esp.: how does your average hourly payment compare to the minimum wage paid in their country of origin)? What was the qualification task?
>
> We recruited crowd-workers via Amazon Mechanical Turk. We didn’t pay by the hour, but by segment, categorized by the number of people in the scene.  These were organized into low-effort (USD150), medium-effort (USD 300), and high-effort (USD 450), corresponding to the amount of estimated time each would take. We estimated the time by comparing it against how long the authors took to annotate segments of each type. We used a reference hourly wage of USD 8 per hour to calculate the reward for each segment, which is slightly above the U.S. minimum wage (USD 7.25). We didn’t explicitly ask for the country of origin of the workers. However, through our close and extended interactions during the annotation period, we did learn of the origin of some of our annotators: India and Brasil. The minimum wages are as follows: India - USD 1.47, and Brasil - USD 5.36.
>
> The qualification task consisted in the annotation of 6 keypoints for 2 participants for 20s of video, and planned to take less than 5 minutes to complete. We paid annotators USD 0.2 for the qualification task (note that typically requesters do not pay for qualification tasks).
>
>
> > Appendix l. 1014: I would recommend using an institutional email address not bound to a person's name here. That way, you can always forward to the currently responsible person without (having to remember) updating the data sheet.
>
> We agree, and have updated this in the Datasheet in the revision.
>
> > End-User License Agreement (EULA) You require the signee of the EULA to have a permanent position at their institution, effectively requiring even senior researchers employed as PostDocs to run the form by their formal bosses. Could you elaborate on the reasoning behind this?
>
> We acknowledge that this could affect some users of the dataset. However, giving access to a researcher in practice involves giving access to their institution, which would normally store the data for them. Not all institutions adhere to the requirement of non-commercial use of the data under our EULA. Therefore, our due diligence process to grant access takes into account both the requestor and their institution. The requirement for a permanent position was put in place to avoid issues caused by researchers changing jobs and keeping the dataset access, or even taking the data with them. This situation is much more likely to occur for researchers in non-permanent positions.

---

> > ### Comment · Reviewer_JhBB · 2022-08-28
> > **Thank you!**
> >
> > Thank you for addressing my questions in the rebuttal and incorporating many of my suggestions in the paper and the appendix.
> > From your original manuscript, it was not clear to me that your phenomenon of interest was "interactions of an international research community" (rather than "social interactions" more broadly); with this interpretation, using the term "in the wild" indeed makes sense.
> >
> > I also recognize your efforts to address ethical issues, and reviewers' concerns more broadly.
> > In order to make sustainable progress beyond leaderboards, AI/ML practitioners need to be more concerned about the details of the data they are working with, and we need more work on responsible data capturing approaches for "difficult" (e.g., hard-to-obtain, potentially sensitive) data.
> > Your work is a valuable step in this direction, and I have raised my score in support of its publication on this track.

---

> > > ### Author Response · Authors · 2022-08-29
> > > **Thank you!**
> > >
> > > Once again, thanks for the meticulousness in your review, and for taking the time to also look at our other responses. It has been refreshing having actual discourse and meaningful discussion. Also thank you for understanding the purpose and context of the dataset, and evaluating it within this context. Your comments, and those from other reviewers, all significantly helped us improve the work, for which we are grateful.

---

> ### Author Response · Authors · 2022-08-16
> **Addressing Additional Feedback: Part 1**
>
> > However, more generally, the title, which mentions the dataset only, is a bit at odds with your actual contributions.
>
> Thank you for the suggestion! We’ve updated the title in the rebuttal revision.
>
> > How is the data from the 4 cameras that are not shared to preserve privacy used in the dataset generation process?
>
> In the current version we used the elevated side-view cameras internally: we evaluated off-the-shelf keypoint detection methods on them before deciding that manual annotation was needed. For a future version, we plan on using them for inferring the height of the keypoints in the top-down view since all cameras are calibrated.
>
> > How do you catch annotation errors or inaccuracies? Can you explain why you deemed it sufficient to have each segment annotated by one annotator only (e.g., why not 2, + 1 for reconciliation)? Was there any kind of quality control? What would be the impact of uncaught inaccuracies in the annotation process?
>
> We performed pilot studies to establish inter-annotator agreement, and discuss this in detail in our dedicated comment on the matter here: [https://openreview.net/forum?id=CNJQKM5cV2o&noteId=jLZ3MEeY6wh](https://openreview.net/forum?id=CNJQKM5cV2o&noteId=jLZ3MEeY6wh)
>
> Additionally, we performed two kinds of pre- and post-validation to ensure the quality of annotations. First, we carefully selected a pool of crowd-sourced workers via a qualification test consisting in the annotation of a short segment of the data. Workers qualified by achieving high inter-rater agreement with our internal annotations done by the authors on the same segment treated as expert annotations. This small, trusted pool did all the annotations. Post-annotation, our annotation tool was developed to enable replaying the annotations (keypoint, or binary speaking status) together with the video. We randomly selected some of the annotations from each HIT for this kind of check. HITs in which we detected missing or low-quality annotations were returned to the annotators for fixing/completing.
>
> > I'm afraid I don't follow the reasoning in this sentence. (1) What are "interactive social systems"? (2) Why is "studying social human behavior in real-life situations" a "crucial challenge" in their "development"? This might be just a problem of phrasing, but your first sentence is currently one of the least clear sentences in the paper (at least to me).”
>
> Thanks for pointing out the lack of clarity. We refer to interactive social systems as artificially intelligent systems that can understand human social behaviour and assist humans in any aspects of their life that involve social intelligence. In hindsight, the use of ‘interactive social systems’ is confusing and we have renamed these ‘artificial socially intelligent systems’. The need for capturing human social behaviour in real-life situations is related to how someone’s behaviour in a social situation changes as a result of variations in social context. The social situation is influenced by how well they know the other interlocutors, relative social status, potential for obtaining resources of use, common interests, etc. We know already from social science that people’s behaviour towards others changes depending on their perceptions of the situation (L22-23 revision). Therefore we see capturing differing social situations such as conferences to be a gap. We’ve updated the opening in the revision.

---

> ### Author Response · Authors · 2022-08-16
> **Addressing Weaknesses**
>
> Thanks for your time, effort, and detailed and thoughtful comments! We sincerely appreciate the detailed feedback and have incorporated the suggestions to improve our paper.
>
> 1\.
> > The overall ConfLab setup appears to be costly (in terms of the financial, temporal, computational, and human resources needed), which limits the replicability of the setup to well-funded institutions or venues.
>
> We acknowledge the overall point since we are aware of the economic privileges associated with being at a European university compared to different parts of the world. Still, in other ways, our core team has been quite small for a project of this scale: a team of 3 Ph.D. students, 1 post-doc, and 1 professor for the entire collection and annotation. An additional Ph.D. student worked part-time on the keypoints task post data annotation. (We don’t mean to diminish the auxiliary support of everyone in the acknowledgments). In fact, our technological innovations for synchronization and continuous annotations have been in part inventions of necessity in order to stay within an academic budget:
>
> - An off-the-shelf wireless synchronization setup costs USD 7500. Our method reduced this to USD 730 by finding a way to use a reference with lower latency, but one within tolerable limits for behavior research [18, Sec. 5, footnote 2].
> - Collecting annotations with Covfee (our continuous annotation framework) at 60Hz was, per our experiments, 3x less expensive than the traditional approach of annotating frames at just 1Hz or lower followed by interpolation. (Note that this comparison is an underestimate in terms of cost savings: continuous annotations at 60Hz capture complex and nuanced dynamics of social behavior that cannot be reconstructed through interpolating between sparse annotations at lower sample rates.)
> - We’ve open-sourced the Midge (our custom wearable) schematics so that others don’t need to spend on the design and development.
>
> Note that all the equipment costs are one-time expenses; the equipment can be reused for multiple data collections. We’ve updated the cost and carbon footprint discussion in the datasheet with the tabulated itemized costs (Table 5 in the Datasheet).
>
> Beyond these individual cost reductions, the broad vision of our ‘datasets by the community for the community’ ethos (L4, 53, 332) is that hopefully no single research group needs to shoulder the economic burden alone. We may not be able to lower the cost floor for studying international communities in their natural ecology. But we chose this mainstream venue hoping that if the broader idea ends up resonating with enough researchers who either (i) appreciate the challenges of data collection, or (ii) will be using the data eventually, it might garner the momentum needed for conference organizers themselves to start contributing time and money to support efforts of this kind at their venues. We can then start moving towards a corpus of diverse datasets based on the ConfLab template where the cost is amortized across research groups.
>
> 2\.
> > The authors discuss some of the related [potential negative societal impacts] concerns in their datasheet, but not in the main paper.
>
> This is indeed planned to be in the main paper (we’d reserved it for the 10th page we get for the camera-ready version). Since we have the 10th page for the revision, we’ve already moved it back now.
>
>
> 3\.
> > Also, "in-the-wild" is over-advertising… the setup is more like "In-the-Zoo".
>
> Our use of “in-the-wild” is closer to the original sense of the term pertaining to studying a phenomenon in its natural ecology, and related to the notion of ecological validity (see [6, 7, 8]). Since we’re studying interactions of an international research community (phenomenon) in its natural ecology (international conference), we believe the term applies. From this perspective, we are actually reclaiming the term from the way it’s been redefined by (us) computer scientists. Besides, we believe “in the zoo” more accurately applies to setups such as the Panoptic studio [24], where the dialogues and interactions can be unscripted and semi-realistic but occur in a dedicated enclosed rig in the lab.  (We agree with the hyphenation though! We realized this too late and fixed it already for the preprint and the dataset documentation on 4TU.ResearchData, but will need to update again following your title suggestions.)

---

### Official Review · Reviewer_jq5N · 2022-07-25
**ConfLab: A Rich Multimodal Multisensor Dataset of Free-Standing Social Interactions In-the-Wild**

**Rating:** 6
**Confidence:** 3
**Correctness:** I don't have concerns about the corre…
**Clarity:** The paper is well written and easy to…

**Strengths:**

1. Compared to existing datasets, ConfLab involves detailed annotations of 17 keypoints and speak status annotations and hardware synchronized camera and multimodal wearable signals.

2. The paper is well-written and the dataset is well documented, which will benefit the potential users.

3. Privacy and ethical issue are well discussed and the dataset collection process is under strict restrictions.

4. The authors provide reproducibility checklist and well organized reproduction procedures.

**Weaknesses:**

1. The authors claim that pre-trained SOTA methods struggle with a privacy-sensitive top-down perspective of person and keypoints detection, which motivates this application. I am a little bit doubtful about the usefulness of this application from a top-down perspective.

**Additional Feedback:**

No additional feedbacks.

**Documentation:**

Yes. The dataset is well documented.

**Ethics:**

Thee dataset design and process were approved by both, the Human Research Ethics Committee at our institution (TUDelft) and the conference location’s national authorities (France).

**Relation To Prior Work:**

The authors carefully and comprehensively discussed the prior works.

**Summary And Contributions:**

In this paper, the authors propose a multisensor dataset to understand human social interactions. This dataset is collected using aerial-view cameras capturing the interactions of attendees during the conference. The dataset supports applications including person and keypoints detection, speaking status detection and F-formation detection.

---

> ### Author Response · Authors · 2022-08-16
> **Addressing Weaknesses**
>
> Thank you for your comments, reviewer jq5N!
>
> > The authors claim that pre-trained SOTA methods struggle with a privacy-sensitive top-down perspective of person and keypoints detection, which motivates this application. I am a little bit doubtful about the usefulness of this application from a top-down perspective.
>
> We note that computer vision researchers who had different motivations than studying social interactions have also independently been interested in such top-down video perspectives in real-life settings:
>
> - The top-down perspective has been commonly used in airport security surveillance tasks (e.g. [Ref 1, Ref 2, Ref 3] below)
> - Since such perspectives are used in crowded scenes, they’ve also been of interest in
> tasks related to filling in occluded keypoints (e.g. [15]).
>
> Moreover, note that the poor performance of SOTA methods was not the only motivation for this work. If the confusion resulted from our description in the second paragraph of the introduction, we have rewritten this in the revised submission to improve clarity (please see the changes that have been highlighted). We meant to establish the intersecting challenges involved in collecting such a dataset.  Given that we need to use aerial views in crowded settings to preserve privacy, we simply observed that this leads to another challenge in obtaining articulated pose annotations, since state-of-the-art keypoint detection models find these views challenging (Fig. 3).
>
> For a more detailed discussion, we request the reviewer to see our common clarification on the main motivations for the dataset and the various considerations that make the top-down perspective an ideal one for such settings: ([https://openreview.net/forum?id=CNJQKM5cV2o&noteId=kxbOf1oGXu](https://openreview.net/forum?id=CNJQKM5cV2o&noteId=kxbOf1oGXu)). Further, in the ‘Motivation’ and ‘Uses’ sections of the Datasheet we point out several useful tasks beyond keypoint detection that our contribution in this work enables.
>
>
> [Ref 1] Real-Time Airport Security Checkpoint Surveillance Using a Camera Network - Wu et al. [https://citeseerx.ist.psu.edu/viewdoc/download?doi=10.1.1.298.981&rep=rep1&type=pdf](https://citeseerx.ist.psu.edu/viewdoc/download?doi=10.1.1.298.981&rep=rep1&type=pdf)
>
> [Ref 2] Correlating Belongings with Passengers in a Simulated Airport Security Checkpoint. - Islam et al. [https://sites.ecse.rpi.edu/~rjradke/papers/islam-icdsc18.pdf](https://sites.ecse.rpi.edu/~rjradke/papers/islam-icdsc18.pdf)
>
> [Ref 3] Real-World Re-Identification in an Airport Camera Network - Li et al.

---

> > ### Comment · Reviewer_jq5N · 2022-08-29
> > **Thank you for the response.**
> >
> > I would like to thank the authors for their thorough responses. I have no further comments.

---

> > > ### Author Response · Authors · 2022-08-29
> > > **Thank you!**
> > >
> > > Thank you for your response, reviewer jq5N! We're glad you found our responses to be thorough!
> > >
> > > We noticed that your score is unchanged, even though you stated you have no further comments. In the last few hours before the deadline, can we do anything to address any remaining concerns that might be preventing a score update? (We're not sure if the final decision is based on the score alone, and to what extent the content of these discussions might be considered).

---

### Official Review · Reviewer_4w88 · 2022-07-26
**A Multimodal Multisensor Dataset of Social Interactions**

**Rating:** 7
**Confidence:** 2
**Clarity:** Yes, the paper is overall well writte…

**Strengths:**

[S1] ConfLab captures high-quality datasets (body keypoints in high sampling rates) and enables multi-sensor data from different modalities. I can sense that the authors put a lot of effort into constructing this dataset.

[S2] The discussion of the relevant methods is thorough. The comparison in Table 1 shows that this dataset has merit.

**Weaknesses:**

In addition to the detection of speech states, I wonder if the authors could provide more benchmarks for this multimodal setup. I think the potential of this dataset in multimodal tasks based on social interaction (detection, prediction) is of interest.

**Additional Feedback:**

N/A

**Correctness:**

The construction of the dataset and evaluating methods are technically sound. The description of data acquisition in the paper is detailed.

**Documentation:**

The documentation seems sufficient and looks good. I had a cursory inspection of the code and it was generally readable.

**Ethics:**

No. This paper emphasizes the preservation of the participants’ privacy

**Relation To Prior Work:**

The discussion about prior works are clear.

**Summary And Contributions:**

This paper introduces a dataset focusing on free-standing interactions in an international conference. To address the problem of capturing data such as pose annotations with extensive social interactions, the authors propose a data collection design named ConfLab.

---

> ### Author Response · Authors · 2022-08-16
> **Addressing Weaknesses**
>
> Thank you for your time and comments reviewer 4w88!
>
> > In addition to the detection of speech states, I wonder if the authors could provide more benchmarks for this multimodal setup. I think the potential of this dataset in multimodal tasks based on social interaction (detection, prediction) is of interest.
>
> Indeed, there are many multimodal tasks that could be done with the data (see the Uses section in the datasheet). We focused on tasks based on their social relevance and familiarity for researchers (potential users) to better situate the uses of the dataset. The keypoint detection and F-formation detection have been most commonly performed as vision-only tasks. Meanwhile, no-audio speaking status detection has been done multimodally at the Mediaeval challenge for three years (2018, 2019, 2020).
>
> Note that other multimodal approaches for the presented tasks do exist, but they are a minority (e.g. [11], and the more recent "Conversational group detection with graph neural networks." - Thomson et al.) The multimodal streams we capture (IMU, proximity) do enable these approaches in the future, as we acknowledge in the Uses section of the datasheet.

---

### Official Review · Reviewer_PsHE · 2022-07-27
**Multiple technical advances, but limited by data quality**

**Rating:** 5
**Confidence:** 3
**Correctness:** See above -- I have reservations abou…

**Strengths:**

This manuscript demonstrates a new technical approach to a complex, multimodal machine learning problem. The data collection contains multiple innovations and the benchmark experiments included span multiple dicsiplines. There is a detailed methodological supplement and ethical discussion.

**Weaknesses:**

The weaknesses of the manuscript are in the quality of the data annotations. The manuscript clarity could also be improved:

(1)	Quality of the video tracking. The authors’ motivate using overhead views to preserve privacy, but ultimately this leads to poor pose estimation, with up to 50% occlusion for each keypoint. This occlusion makes interpretation of the following results difficult. If part of the motivation of the data is as use for keypoint estimation from overhead video, then I would hope the quality of the ground truth would be higher, eg including ground truth data from the side view cameras, multiple annotator labels, estimates of MPJE etc.

Because of the occlusions, much of the data is imperfect or even removed (‘L 1079 in the supplement If there are more than 50% keypoints missing for a person, we remove the person bounding box and keypoints from the ground-truth.) As a result the utility of having in the wild data has to be balanced against the fact that the data is missing salient events. This issue affects all downstream tasks as well, e.g. for F-formations it is unclear how much the accuracy is impaired by noisy or missing keypoints.

Overall, it seems like a superior approach would be to conduct either semi-natural recordings in a lab with superior ground truth, or to use side view cameras to do the pose annotations but only release the overhead if privacy is truly the concern. It isn’t clear how the side-view cameras are used.

(2)	Quality of annotations and inter-rater reliability. It is difficult to interpret the efficacy of some of these ML approaches, because there is no sense of the inter-annotator variability to gauge performance. Is 80% precision in vocal turn taking good? It is hard to assess without understanding what fraction arises from annotator variability. It also seems the audio data would be useful here for labeling? Similar holds to an ever larger degree for keypoint data as mentioned above and for F-formations.

Minor points:
Minor
(3)	In the supplement it mentions the cameras were calibrated, but I don’t see any use of 3D pose data.
(4)	At some times there were 8 cameras mentioned, other times 14, but only a subset were analyzed.
(5)	I found the description of how different camera views were reasoned across for F-formation detection confusing.
(6)	The authors emphasize that the data was synchronized and thus could be analyzed at high framerates, but the speaking status is only annotated at 1 Hz and I believe the F-formations
(7)	Use of low-frequency audio – was this data used in any of the experiments? Or the Bluetooth proximity data? It seems relevant to the audio turn taking tasks and F-formation respectively.
(8)	There is lots of tangential information that could be removed eg: L200, context of using the mobile phone, L283 discussion of conversation floors
(9)	How were identities from the midge associated with visual keypoint data?
(10)	Elaborate on definitions for F-detection, as this is a relatively specialized topic that may not be familiar to many readers.
(11)	Table 1 – is the dataset 45 minutes long or 15 minutes from multiple modalities?
L294 – development typo


**Additional Feedback:**

See above.

**Clarity:**

The clarity is okay. Overall the manuscript felt a bit unfocused, in part because it is trying to do so many things (a new dataset, a new dataset concept, multiple technical advances as well as benchmarks, across several different fields). eg	There is lots of tangential information that could be removed eg: L200, context of using the mobile phone, L283 discussion of conversation floors.

I would suggest streamlining the manuscript, removing the discussion of the new dataset collection concept.



**Documentation:**

The documentation is sufficient.

**Ethics:**

EThics are a strength of this manuscript and are well accounted for.

**Relation To Prior Work:**

Yes.

**Summary And Contributions:**

This paper introduces the Conflab dataset, a multi-modal dataset of social interactions in-the-wild. Data from 15-45 minutes of 48 individuals interacting at a conference from 10 overhead cameras, as well as a badge-borne accelerometer and audio recordings. Multiple hardware advances are made to allow for synchronization across cameras and devices, as well as inclusion of multiple datastreams on the badge. The authors’ also collect survey data on participants with information about their senority and conference interests. The raw data is annotated with body keypoints, speaking status, and F-formation identity, a form of social grouping. Several experimeriments are performed on (1) keypoint estimation from overhead cameras (2) turn-taking from keypoint and accelerometer data (3) F-formations detection from different keypoint subsets. The dataset is well documented, with a complete datasheet and discussion of privacy.

This manuscript describes a problem that I feel is of broad interest to the community, and I believe describes multiple novel advances. However I think the manuscript is undermined as a dataset by the quality of the video tracking and somewhat cursory annotations of the data described below. Because of these weaknesses, I am not sure that it will find broad utility in the field, and as a result may be better suited to a specialized journal.

---

> ### Author Response · Authors · 2022-08-16
> **Addressing Minor Points and Clarity Comments**
>
> 3\.
> > In the supplement it mentions the cameras were calibrated, but I don’t see any use of 3D pose data.
>
> We do not use 3D pose data in our experiments. We aligned with the common behavior analysis pipeline which uses 2D keypoints detected in video for skeleton-based action recognition.
>
> This does not preclude future efforts with 3D pose data. The extrinsics of the elevated side views allows for estimating the height of the keypoints in the top-down cameras even for those keypoints occluded in the side-views. This is something we plan on releasing in the future.  Additionally, calibrating cameras allow for obtaining correspondences across overlapping camera views.
>
> 4\.
> > At some times there were 8 cameras mentioned, other times 14, but only a subset were analysed.
>
> We used 10 top-down and 4 elevated side-view cameras (not shared due to privacy) (revision L134-137). Due to the heavy overlap between the top-down views, we annotated from cameras 2,4,6,8, and 10 (which already provide complete coverage), so there is no loss of information (See Fig. 1). We state that Camera 5 failed 30 minutes into the event (Fig.1 caption, Datasheet Composition section).
>
> 5\.
> > I found the description of how different camera views were reasoned across for F-formation detection confusing
>
> F-formations (groups) can span across different camera views. While annotating, we always picked a camera view that captures each F-formation in its entirety.
> We have clarified this in the main text.
>
> 6\.
> > The authors emphasise that the data was synchronised and thus could be analysed at high framerates, but the speaking status is only annotated at 1 Hz and I believe the F-formations
>
> There appears to be a misunderstanding on this point. Our videos were recorded and annotated at 60Hz, including both poses and speaking status (see Table 1).
>
> Only F-formations were annotated at 1Hz. Even Kendon’s original discussions are only well defined at rates of <1Hz. This is also why they were annotated in previous datasets at 1Hz [9], or lower frequencies [21, 23].
>
> 7\.
> > Use of low-frequency audio – was this data used in any of the experiments? Or the Bluetooth proximity data? It seems relevant to the audio turn taking tasks and F-formation respectively
>
>
> Because of the privacy challenges around the collection of audio (even low-frequency), social interaction datasets like Conflab normally have no raw audio (Table 1). This was our motivation for addressing the task of no-audio speaking status detection (revision L268), and the reason we did not use audio as input in the baselines.
>
> Similarly, we chose to benchmark F-formation against SOTA methods with visually obtained information (i.e., using locations and orientations) since it is the mainstream approach.
> However, our collection of low-frequency audio and proximity data allows a rich set of future studies looking into the quality of speaking annotations using audio or multimodal data.
>
>
> 8\.
> > There is lots of tangential information that could be removed eg: L200, the context of using the mobile phone, L283 discussion of conversation floors
>
> How to deal with mobile users in relation to associates of F-formations is still an open question, so the annotations are useful. However, we agree it is misplaced in the text and have moved this to the Datasheet (Uses).
>
> The explanation of conversation floors is important to interpret the results, however. We’ve made the text clearer in the revision.
>
> 9\.
> > How were identities from the midge associated with visual keypoint data?
>
> We discuss data association in L151-160 (revision). We designed a registration protocol at the event where we took a picture of each participant with their Midge. Annotators were presented with a gallery of representative images of people under the camera and their corresponding midge ids (see Fig. 6b).
>
> 10\.
> > Elaborate on definitions for F-detection, as this is a relatively specialized topic that may not be familiar to many readers.
>
> Thanks; we have added a definition in L210-214 in the revision.
>
> 11\.
> > Is the dataset 45 minutes long or 15 minutes from multiple modalities?
>
> The raw data (videos and wearable signals) are 45 minutes long. Of these, 16 minutes were annotated with keypoints, F-formations, and speaking status.
>
> > I would suggest … removing the discussion of the new dataset collection concept.
>
> The prohibitive cost and effort to record at a conference is often underestimated by the community. Our ‘datasets by the community for the community’ ethos (L4, 47) is therefore important for a joint effort to study such social behavior in the natural ecology (also see our response for the cost-related comment by JhBB: [https://openreview.net/forum?id=CNJQKM5cV2o&noteId=yPNziDRBez7](https://openreview.net/forum?id=CNJQKM5cV2o&noteId=yPNziDRBez7)). Our vision is that the ConfLab template can be used by multiple groups towards building a corpus of similar datasets across scenarios to enable new research questions.

---

> > ### Comment · Reviewer_PsHE · 2022-08-28
> > **Thank you for the clarifications**
> >
> > Thank you for the clarifications. I very much appreciate the clarification of occlusions and keypoint accuracy, and the enclosed video. Can you clarify whether the enclosed approach was part of the training or test set for keypoint evaluation, ie to what degree this is representative of overall dataset quality? In light of this I will increase the score to a 5.
> >
> > Overall, I still have reservations about the quality of this dataset for driving novel AI/ML approaches in the neurips community. I see the ideal dataset as part of this track as being one where AI practicioners can make progress without having to be overly concerned with the details of the dataset. While the philosophy and approach driving this effort is potentially valuable, it is very difficult to set up and collect data in the wild, and comes with a loss of quality and control and scale. This is not a statement against in the wild approaches in general, more against this one in particular. It is on the smaller end, the quality is a bit mixed, the ability to set up well-controlled train/test/val splits in light of this variation can be challenging. People are out of frame, the number of people in the field is variable, the audio didn't work, the 2D views suffer from large distortions that are uncorrected which might make it difficult to generalize the keypoint tracking etc.  I respect that other reviewers closer to the field may still see this matter differently.

---

> > > ### Author Response · Authors · 2022-08-29
> > > **Clarifications surrounding in-the-wild approaches, and broader meta-discussion**
> > >
> > > ### Clarification 5 : What we mean by ‘in-the-wild’
> > >
> > > There is a potential misunderstanding of the reviewer with respect to the term ‘in-the-wild’ which may be best clarified by referring to the discussion with reviewer JhBB on the difference between ecological validity (point 3: [https://openreview.net/forum?id=CNJQKM5cV2o&noteId=yPNziDRBez7](https://openreview.net/forum?id=CNJQKM5cV2o&noteId=yPNziDRBez7)). Based on our reclaiming of the term ‘in the wild’ to better reflect ecologically valid data collection, we would argue that the “larger” “high quality” multimodal datasets that the reviewer might be referring to are predominantly made from scraping data off the internet which is not comparable to the types of data related to this work (e.g., COCO dataset, and CMU MOSI and MOSEI datasets). Our dataset, even in this first instance is higher quality and of a comparable size to any other state-of-the-art related dataset.
> > >
> > >
> > > ### Broader meta-discussion
> > >
> > > We recognise PsHE’s comment to be representative of a broad cross-section of the community and very much embedded in the historical ethos. However, with global trends calling for AI to become more ethically aware, one can no longer argue that data can be considered a commodity independent from the ML task at hand. As JhBB points out: ` In order to make sustainable progress beyond leaderboards, AI/ML practitioners need to be more concerned about the details of the data they are working with, and we need more work on responsible data capturing approaches for "difficult" (e.g., hard-to-obtain, potentially sensitive) data.` ([https://openreview.net/forum?id=CNJQKM5cV2o&noteId=m2d4jQDSzYe](https://openreview.net/forum?id=CNJQKM5cV2o&noteId=m2d4jQDSzYe))
> > >
> > > As Social AI is emerging, we see significant inertia in pushing the field truly forward. The entrenched ML viewpoint on what good quality data should be forgets that good quality data should be collected in partnership with the nature of the problem that is to be solved.
> > >
> > > To illustrate this point at a broader level, let us take a perhaps more inane example. Let us consider object detection, which historically would have seemed to be purely an objective perception task. However, this misses the cultural and social history that brought those objects into being in the first place. Even discussions of objects as socio-economic and cultural artefacts seem to only make it as side discussions in workshops at ML conferences rather than important issues to put center stage [Ref1].
> > >
> > > Note that as the popularity of the metaverse takes flight, we have heard of researchers being disappointed that pretrained object detection modules do not work well when used with the oculus rift in someone’s home kitchen. This is because there is a severe lack of representative cluttered (or occluded) data. Current pretrained SotA object detectors are trained on data that does not fit the ecology of the situation; simply put, the “clutter” in someone’s kitchen is not the same as the “clutter” observed in scraped online data that is typical of stock footage; objects are co-located in a person’s kitchen due to a particular individual’s lifestyle, causing occlusions that might be uncharacteristic of stock photo images where the popular layout of kitchen items might be considered [Ref 2, Ref 3].
> > >
> > > If we transition this argument back towards ConfLab, the organization of people and groups dictates behavior that is unique to that group in that context. Collecting appropriate data in the right context is hard. But it is an *absolutely necessary part of the task of understanding what and how to adapt to specific situations*. Our work here constitutes the first step towards this end; therefore we believe that the idea of the collection concept itself is also an important contribution.
> > >
> > > While this discussion may seem to be orthogonal to the point of the reviewer’s comment, we argue that the main reason for bringing this up is to showcase how the ML community in general still finds it less relevant to consider collecting data designed for people moving around their every day lives;  It has been thus far, much easier to scrape data from the internet and consider this to be sufficiently in the wild and true enough to real life. We argue against this if the field, in general, is going to move forward; AI practitioners ought to still be concerned with the details of the data they use.
> > >
> > > [Ref 1] Does Object Recognition work for everyone? ( https://arxiv.org/abs/1906.02659)
> > > [Ref 2] Thing Ethnography: Doing Design research with Non-humans (https://dl.acm.org/doi/10.1145/2901790.2901905)
> > > [Ref 3] Aspects of Everyday Design: Resourcefulness, Adaptation, and Emergence (http://dx.doi.org/10.1080/10447310802142276)

---

> > > ### Author Response · Authors · 2022-08-29
> > > **Factual clarifications about audio and cameras**
> > >
> > > ### Clarification 3: “Audio didn’t work”; actually it does
> > >
> > > Please note that this is an incorrect interpretation. The audio does work: we clarified this in our response ([https://openreview.net/forum?id=CNJQKM5cV2o&noteId=jLZ3MEeY6wh](https://openreview.net/forum?id=CNJQKM5cV2o&noteId=jLZ3MEeY6wh)). Additionally, we have included a report on the intelligibility of words in the low-frequency audio data as part of our due diligence on privacy preservation of the audio data. (This is in the newest Supplementary revision for convenient access, will be uploaded online later.)
> > >
> > > If the reviewer is referring to why we did not use the audio for the baseline tasks. Even if we had had it, we would not have used it in this paper to generate the ground truth. This is due to the following reasons:
> > >
> > > - From inspection of the low frequency audio data, it is not always obvious when someone is speaking, particularly if the voice of the person wearing the midge is softer than those of their interlocutors. So manual annotation using the audio data would not be appropriate.
> > > - Moreover, SoTA Voice activity detectors cannot work reliably out of the box on low frequency audio data. We have run adapted versions on a subsampled high frequency audio dataset that we have in-house that has much better SNR. The results were not sufficiently robust for us to trust an automated reference generation for our noisier low frequency audio data.
> > > - We concluded that perceiving the ground truth speaking activity multimodally could be more appropriate based on the pilot study. However, how this relates to video-only or audio-only annotations of speaking status and how this affects model training is unknown. We felt it would not be appropriate to just provide multimodal annotations without a detailed analysis of how they differ from audio only and video only annotations. To link back to existing work, we rely on approaches that use video-only annotation as the benchmark test.
> > > - Comparison with an A/V or audio only reference would be meaningless without a further analysis of exactly how onsets and offsets of speech activity differ. The nature of the problem requires further investigation. Sticking with prior approaches was the only way to make the work comparable.
> > >
> > > ### Clarification 4: People out of camera, variable number of people in the field, generalizing knowledge to other settings
> > >
> > > The reason we marked out the floor grid to compute extrinsics is so that the views can be combined into a single view of the entire scene if needed. This is what we meant by correspondences across views in point 3 in our response here: [https://openreview.net/forum?id=CNJQKM5cV2o&noteId=eKNkEMTHGwX](https://openreview.net/forum?id=CNJQKM5cV2o&noteId=eKNkEMTHGwX). (We plan on releasing such a version in the future.)
> > >
> > > However, we believe partial people in the frame and varying densities per camera does not undermine quality, nor does it invalidate the experiments. We actually consider having variable people density a positive attribute. These attributes, along with our one-camera-at-a-time experimental setup with people moving across top-down cameras matches real-world use cases even beyond social settings. An example real-world setting pertains to airport security; we’ve referred to papers studying these settings in our response here: [https://openreview.net/forum?id=CNJQKM5cV2o&noteId=vBrHDULT3Pb](https://openreview.net/forum?id=CNJQKM5cV2o&noteId=vBrHDULT3Pb)

---

> > > ### Author Response · Authors · 2022-08-29
> > > **Thank you! Some factual clarifications, and meta-discussion about contributions and evaluation of dataset construction methodology**
> > >
> > > Thank you for your response, reviewer PsHE! We address the clarifications before engaging in a broader discussion about what this dataset means for in-the-wild behavior research.
> > >
> > >
> > > ### Clarification 1: Representativeness of the visualization of overall annotation quality
> > >
> > > The annotation visualization is representative of the overall data. No additional data processing was done in creating them. We simply plot the released keypoints onto the images from the available data. (Any user would be able to construct the visualization as is). No distinguishing treatment was done while annotating different segments either.
> > >
> > > > Can you clarify whether the enclosed approach was part of the training or test set for keypoint evaluation
> > >
> > > By enclosed approach, we believe you mean the pre-filtering in the keypoints experiments of people at the edges if more than half their body is off camera. This was done as a global preprocessing before generating splits, and affects all splits (train/val/test) (L1132-1133). We believe this is in fact the correct setup since it ensures similar conditions across splits.
> > >
> > > ### Clarification 2: ConfLab as a data collection concept is valid wrt the CFP
> > >
> > > > I see the ideal dataset as part of this track as being one where AI practicioners can make progress without having to be overly concerned with the details of the dataset
> > >
> > > Perhaps the reviewer is viewing in-the-wild image datasets for keypoint detection (such as MSCoco) as a frame of reference to evaluate this dataset. We believe a more appropriate frame of reference the state of in-the-wild behavior datasets (Also see our comment below pertaining to the term 'in-the-wild': [https://openreview.net/forum?id=CNJQKM5cV2o&noteId=m0HT1pNdL2w](https://openreview.net/forum?id=CNJQKM5cV2o&noteId=m0HT1pNdL2w)). Given that in the original review the reviewer suggested `streamlining the manuscript, removing the discussion of the new dataset collection concept`, it appears that there is a misalignment on how we and the reviewer interpret the goals of this track.
> > >
> > > From the CfP and the blog post about why this track was started:
> > >
> > > > The Datasets and Benchmarks track serves as a novel venue for … a forum for discussions on how to improve dataset development… Submissions to this track will be reviewed according to a set of criteria and best practices specifically designed for datasets and benchmarks
> > >
> > > > Hence, many researchers resort to data that are conveniently available, *but not representative of real applications*. For instance, many algorithms are only evaluated on toy problems, or data that is plagued with bias, which could lead to biased models or misleading results, and subsequent public criticism of the field (Paullada et al. 2020).
> > >
> > > In this work, the application pertains to understanding in-the-wild behavior. Pertaining to this application, we have significantly advanced the techniques and considerations for collecting behavior data in natural ecologies over the existing state of the art. The data collection concept is therefore crucial to enable the collection of a corpus of such datasets across conferences. Note that a single such effort incurs significant labor and cost, so a reproducible method for similar data collections is important. Please see point 1 in our response here: https://openreview.net/forum?id=CNJQKM5cV2o&noteId=yPNziDRBez7](https://openreview.net/forum?id=CNJQKM5cV2o&noteId=yPNziDRBez7.

---

> ### Author Response · Authors · 2022-08-16
> **Clarifying inter-rater reliability and interpreting efficacy of methods**
>
> 2\.
> > It is difficult to interpret the efficacy of some of these ML approaches, because there is no sense of the inter-annotator variability to gauge performance. Is 80% precision in vocal turn taking good? It is hard to assess without understanding what fraction arises from annotator variability. It also seems the audio data would be useful here for labelling? Similar holds to an ever larger degree for keypoint data as mentioned above and for F-formations.
>
> We agree that inter-rater reliability estimates will help readers interpret the data. In pilot studies preceding the annotation of the complete dataset we calculated these estimates on a smaller set of annotators.
>
> ### Keypoints
>
> For keypoints, we performed a pilot evaluation comparing accuracies for our continuous annotation against the established traditional method of annotating keypoints with interpolation using the CVAT tool [46]. In a pilot with 3 annotators in each condition (tool), we found no statistically significant differences in errors per-frame, despite a 3x speed-up in annotation time. We have made this more clear in revision L185-187. Please refer to the annotation visualization video for a qualitative impression of errors.
>
> ### Speaking Status
>
> Regarding speaking status annotation, the question of how to measure subjectivity was intensely discussed during writing. No standard agreement metric exists to account for subjectivity in binary time series data; in a pilot with 2 annotators, we measured a frame-level agreement (Krippendorf’s alpha) of 0.552. This is in line with previous work [9] which annotated speaking status and other social actions. Note that [9] reports inter-annotator agreement using Fleiss Kappa (exactly equivalent to Krippendorff’s alpha in this case) of about 0.5 for speaking (exact numbers are not provided, see [9, Fig.8] ). Reference [42] ( a widely used dataset of dyadic conversations) reports lower agreements in the range 0.17-0.31 (Cohen’s Kappa) for five dimensions of social behaviour. We have included this discussion in the Datasheet (Composition section, Q.2) and the main paper (revision L201-205).
>
> ### F-formations
> Our annotations of F-formations are strictly based on Adam Kendon's original definition which is a largely objective definition of identifying people in an arrangement depending on their facing direction. Specifically, in Kendon’s definition, F-formation membership is measured in terms of the location and orientation of the lower body. When there is no longer an overlap between the 90 degree frustum extending in front of each person, the F-formation is ended. This framework removes most ambiguities in how to identify F-formations. While it could be worthwhile and interesting to measure agreement on F-formation group membership, it is likely that it will result in high agreement if all annotations follow this objective definition.
>
> ### Interpreting efficacy of ML approaches
>
> Regarding the interpretation of performance, however, we believe that comparing with previous work may be more informative . In detecting speaking status from acceleration, we followed the same data processing and evaluation setup as in [17], but obtained AUCs of around 0.8, compared to 0.68 (mean) in this previous paper.
>
> ### Why audio was not used for labelling speaking status
>
> Audio data was not used for annotation due to the hardware malfunction mentioned in the datasheet section Composition Q.2. We have since resolved the issue by manually aligning the audio data. However, we do not have audio only annotations yet: it is nontrivial to obtain robust voice activity estimates at 1250Hz using an off-the-shelf voice activity detector. A manual annotation process will be required on the same scale as the video-only annotations. For now, we chose to follow the video-only annotation procedure which has proved to provide useful supervisory signals [9].

---

> ### Author Response · Authors · 2022-08-16
> **Clarifications: Camera perspective choice, occlusion, and annotation quality of keypoints**
>
> Thank you for your time, comments, and feedback reviewer PsHE! We sincerely appreciate the detailed comments that has allowed us to improve the paper in the revision.
>
> We believe that there might be a potential misunderstanding surrounding the quality of the keypoints annotations. We clarify the details below and have update the confusing text in the revision. Moreover, for qualitative context, we’ve added a visualization of the annotations for a data segment of approximately 2.5 min with dynamic salient events to the revised supplementary material (`keypoints_annotations_sample.mp4`, which will also be added to the samples directory in the dataset on 4TU.ResearchData). We request the reviewer to kindly see the video in addition to our responses below.
>
> 1a\.
> > Because of the occlusions, much of the data is imperfect or even removed (‘L 1079 in the supplement If there are more than 50% keypoints missing for a person, we remove the person bounding box and keypoints from the ground-truth.)
>
> We now see that we should have been clearer in L1079 and have updated the text in Appendix E.1 in the rebuttal revision to address the discussion below.
>
> The misunderstanding is that the data was not removed due to poor quality or occlusion, i.e., body parts blocked by other people or other body parts. Rather, what we are referring to are people at the edge of a camera’s field of view that might be partially out of frame (see the top of Fig. 6a). For the person detection task, we compute the bounding box from the keypoint ground-truth annotations. If more than half the body is out of the frame (50% keypoints) so that e.g. only their legs are visible (revision Fig. 7a top), we don’t consider the person for that frame in the experiments. But note that due to the significant overlap between the camera views (Fig. 1),  they would be considered for the corresponding frame in the next camera. If they move back, they are again taken into consideration for the original camera.
>
> Also note that the occlusion value is only a flag to denote that the keypoint is not directly visible. We asked the annotators to still label the occluded keypoints using the best estimate if the point was deemed to be within the frame. These are visible in the annotation visualisation.
>
> 1b\. *Further clarification about the occlusion numbers in Fig. 6c (now 7c in the revision)*
>
>
> Please note that the occlusion values in Figure 6c were conservative, and potentially inflated: we computed them per frame, counting all off-camera keypoints for a person at the edge as occluded, even if they are visible in another camera. We recomputed the numbers by correcting for this fact in the revision, and the resulting occlusion values are 10-25% lower.
>
> 1c\.
> >   [T]he utility of having in the wild data has to be balanced against the fact that the data is missing salient events. This issue affects all downstream tasks as well, e.g. for F-formations it is unclear how much the accuracy is impaired by noisy or missing keypoints.
>
>
> If the reviewer meant that we’re missing important information due to the misunderstanding that 50% of our keypoint annotations were dropped, hopefully the previous clarification addresses this issue. As seen in the visualization video, the annotated keypoints do capture salient gestures and movements of people across the interaction area.
>
> 1d\.
> > The authors’ motivate using overhead views to preserve privacy, but ultimately this leads to poor pose estimation, with up to 50% occlusion for each keypoint… Overall, it seems like a superior approach would be…  to use side view cameras to do the pose annotations.
>
> In fact, the elevated side-view is in fact **worse** for occlusion issues compared to the top-down perspective. We request the reviewer to kindly refer to the ‘Technical considerations’ part of our common clarification ([https://openreview.net/forum?id=CNJQKM5cV2o&noteId=kxbOf1oGXu](https://openreview.net/forum?id=CNJQKM5cV2o&noteId=kxbOf1oGXu)) for a detailed discussion.
>
> In L33 (old version), we have used ‘aerial perspective’ to include both our elevated side-view and top-down perspectives. Both are challenging for state-of-the-art methods to obtain automatic keypoint annotations: the perspective in [15] referenced in L35 (old version) is actually the elevated side-view, see [15, Fig. 1, 2, 7]. We noticed the confusion arising from L35 (old version) and rewritten that paragraph in the introduction to make things clearer.

---

> ### Author Response · Authors · 2022-08-27
> **Requesting response to potentially further address any remaining concerns prior to the discussion deadline**
>
> Hi Reviewer PsHE!
>
> We've tried to address all your points across our responses here and the common clarification comment. Crucially, as we've mentioned, we think there is a possibility that the comments about annotation quality might have arisen from a misunderstanding. (We've added a visualization video to the revised supplementary.)
>
> Since we only have two more days before the discussion deadline, we are hoping to get your response soon to see if this is the case, so that we have a chance to address any remaining concerns or open questions before it is out of our hands.
>
> Thank you for your time in engaging with our paper!
>
> Best,
>
> Authors

---

### Official Review · Reviewer_XnKf · 2022-07-28
**Useful dataset for understanding social interactions**

**Rating:** 7
**Confidence:** 4
**Correctness:** Yes
**Clarity:** Yes

**Strengths:**

- The paper provides a comprehensive description of how data is collected, prepared and benchmarked for multiple tasks.
- It involves a tremendous amount of work for visual annotations (if done properly).
- The dataset includes a higher number of subjects, better video quality, and wearables signals with wireless hardware synchronisation.
- Both individual-level and group-level statistics were provided.
- Useful for many applications in a broader context.
- Privacy preserved for human subjects.
- Interesting and challenging for SOTA.

**Weaknesses:**

- Limited emotional expressions in social interactions.
- Lack of benchmarking results on recent SOTA methods as the paper only included very limited number of methods in the benchmark
- Lack of background and context diversities


**Additional Feedback:**

NIL

**Documentation:**

Well-documented

**Ethics:**

Declared in page 4.

**Relation To Prior Work:**

The paper included a comprehensive comparison with prior datasets of free-standing people in the wild.

**Summary And Contributions:**

The paper proposes a dataset with human interactions recorded in an ACM MM workshop. It involves multiple overhead videos, audio and wearable sensors with privacy-preserving. The author group has also provided continuous key points, speaking statuses and F-formation annotations. The dataset support person/keypoint detection, speaking status detection, and F-formation detection. In a broader context, this dataset can be useful for many applications such as security surveillance and social signal processing.

---

> ### Author Response · Authors · 2022-08-16
> **Addressing Weaknesses**
>
> Thank you for your time, and the thoughtful comments about our work!
>
> 1\.
> > Limited emotional expressions in social interactions.
>
> If the reviewer is referring to *facial* expressions, capturing faces was an extended topic of discussion while selecting the camera perspective. We also discussed this with the Human Resources Ethics Committee at TUDelft and the General Chairs of Multimedia 2019. We concluded that the risk of including faces towards enabling potentially nefarious downstream applications is too significant. (This is also why we were not allowed to share videos from the elevated side-view cameras) We request the reviewer to please refer to our Clarification comment about the choice of camera perspective ([https://openreview.net/forum?id=CNJQKM5cV2o&noteId=kxbOf1oGXu](https://openreview.net/forum?id=CNJQKM5cV2o&noteId=kxbOf1oGXu)), where we further discuss the risks under “Ethical Considerations”.
>
>
> In any case, we do indeed capture the emotional expressions through the body language and gestures through our continuous keypoint annotations. Note that no prior in-the-wild dataset has provided such articulated pose annotations (see Table 1).
>
> 2\.
> > Lack of benchmarking results on recent SOTA methods as the paper only included very limited number of methods in the benchmark
>
> In our benchmarks, we prioritised benchmarking SOTA methods whenever possible. In the F-formation detection task, methods made for a fixed number of people in the scene were not applicable (revision L296). We benchmarked two SOTA F-formation detection models among the ones without this assumption. In the speaking status detection tasks, the four models evaluated are among the best performing in time series classification and skeleton action recognition standard benchmarks (revision L272). In the keypoints detection task, we do provide qualitative results from several pretrained SOTA methods (see Fig.3, Appendix F.1 Fig. 17,18 ). Since we retrain a baseline method on our dataset, we chose to start with a simpler method and used Mask-RCNN as a starting reference to build upon (qualitative comparison in Appendix F.1 Fig. 18).
>
> 3\.
> > Lack of background and context diversities
>
> Ideally, we would also like a dataset capturing multiple such interactions across conferences and events. In reality, the effort for recording on event while adhering to ethical best-practices is often underestimated. Due to the prohibitive requirements (technical, logistic, ethical) for a single such recording, prior related datasets (Tab.~1) also capture a single setting which are still useful. (Also see our response to reviewer JhBB’s comment about the cost: [https://openreview.net/forum?id=CNJQKM5cV2o&noteId=yPNziDRBez7]( https://openreview.net/forum?id=CNJQKM5cV2o&noteId=yPNziDRBez7) ).
>
> In fact, the prohibitive effort involved is precisely why we have argued for a ‘by the community for the community’ ethos for such data collection efforts in the real world (revision L4, 53, 332).  We hope that the existence of this first dataset of this kind would kickstart more widespread involvement of joint efforts across labs and potentially conference organizers themselves to amortize the cost and effort for data collection across multiple groups. We would then be able to augment ConfLab into a significantly larger corpus of more settings, subjects, and backgrounds.

---

> > ### Comment · Reviewer_XnKf · 2022-08-29
> > **Reply to authors**
> >
> > Thank authors for addressing my concerns. It is also good to see your comments on ethical concerns. I am keeping my score. All the best.

---

### Review · Ethics_Reviewer_iFrJ · 2022-08-22

**Recommendation:** 1

**Ethics Documentation:**

Yes, this is explicitly considered in the paper.

**Ethics Review:**

The authors have considered the ethical concerns regarding possible privacy infringement and malicious surveillance from the use of this data and the overarching method. They have taken a number of steps to mitigate these concerns including by finding ways to obscure faces and voice, and ensuring the consent of those participating in the curation of the dataset. This sets up a useful precedent for other work in this space.

However I do have a remaining concern involving the overall goal the paper and the approach. The authors indicate that this work might assist in improving social interaction, but I fail to see how this sort of effort will help actually improve the quality and quantity of human engagement. My worry is that the more likely use of this sort of research is for private companies to sell tools that claim they can help build relationships, but these tools seem more likely to benefit the companies than the actual individuals. What actually beneficial use cases will this type of research be put towards and how does that actually help improve social interaction without attendant harms? Will these companies likely limit themselves to the reasonable restraints the authors have developed, and how can this be assured?

---

> ### Author Response · Authors · 2022-08-24
> **Beneficial use cases and incorporating harm-avoidance into design philosophy**
>
> 2\.
> >  What actually beneficial use cases will this type of research be put towards and how does that actually help improve social interaction without attendant harms?
>
> The social signal processing and human-robot interaction communities have long had the goal of developing systems that improve social experience [Ref 3, Ref 4]. The applications we list below are just some out of many potential use cases.
>
>
> ### Possible application 1: Quantified Self for wellbeing
> One immediately beneficial use-case was for the participants themselves, following our *dataset by the community for the community* ethos. The participant report we shared with the participants after the event ( L113-116, Appendix C for a sample report) provided insights into their own behavior. Such self-tracking or quantified self has been a popular movement for self-empowerment in cases of monitoring and improving one's own health and wellbeing [Ref 1, 2].
>
> ### Possible application 2: Helping to initiate social encounters
> A spin-off of ConfLab called ConfFlow [Ref 8] (deployed at 4 international conferences in 2020-2021) was set up to help researchers navigate the social aspects of scientific discourse and collaboration in the research community. Understanding what aspects of initial encounters lead to people being selected as collaboration partners, and whether these selections are fruitful are part of an ongoing interdisciplinary collaboration in our lab [Ref 9, 10].
>
> ### Possible application 3: Organizational Learning, Employee well being, and Team performance
> The role of free standing conversations in affecting people’s lives goes beyond just social experience in conference networking events, affecting organizational learning [Ref 11] (the process of knowledge transfer within organizations [Ref 5]). Conversations by the water cooler or coffee machine have long been recognized as important moments of organizational learning . The quality of ad-hoc discussions in teams [Ref 12] is also relevant in long term space missions to mitigate team conflict and breakdowns [Ref 14].
>
> ### Harm avoidance: The ConfLab philosophy aims to empower users by taking an agentist vs structurist approach
> The analysis of social behaviour in such settings has classically taken a more top down perspective, e.g. in the social-network analysis domain where the analysis of interactions (via only proximity networks) in conferences has been used to study the process of making science in the field of Meta Science [Ref 6]. However, while social network science is a very well populated domain, it misses a more individualised measurement of social behaviour; see more discussion of the structure vs. agency debate [Ref 7]. Relying on the network science approach jeopardises an individuals’ right to technologies that enable free will, which we consider a form of individual harm avoidance. ConfLab provides access to more than just proximity data about social interactions, enabling context specific social dynamics which vary as a result of an individual’s own context as well as that of the group they are interacting with [Ref 13]. Harm can be further mitigated by bringing awareness of participatory design practices and value-sensitive design principles in the development of assistive technologies.
>
> [Ref 1] Understanding quantified-selfers' practices in collecting and exploring personal data (https://doi.org/10.1145/2556288.2557372)
>
> [Ref 2] What Does All This Data Mean for My Future Mood? (https://doi.org/10.1080/07370024.2016.1277724)
>
> [Ref 3] Bridging the Gap Between Social Animal and Unsocial Machine: A Survey of Social Signal Processing - Vinciarelli et al.
>
> [Ref 4]  What are We Measuring Anyway? (https://doi.org/10.1145/3308532.3329421)
>
> [Ref 5] Productivity Through Coffee Breaks (https://papers.ssrn.com/sol3/papers.cfm?abstract_id=1586375)
>
> [Ref 6] Initiating scientific collaborations across career levels and disciplines (https://doi.org/10.1007/s11412-021-09345-7)
>
> [Ref 7] Free Will, Determinism and the “Problem” of Structure and Agency in the Social Sciences (https://doi.org/10.1177/0048393118814952)
>
> [Ref 8] Encouraging Scientific Collaborations with ConfFlow 2021 (https://records.sigmm.org/2022/04/20/encouraging-scientific-collaborations-with-confflow-2021/)
>
> [Ref 9] Designing Hybrid Intelligence Techniques for Facilitating Collaboration Informed by Social Science, Hrkalovic, to appear ICMI 2022
>
> [Ref 10] A Research Agenda for Hybrid Intelligence (https://DOI: 10.1109/MC.2020.2996587)
>
> [Ref 11] The Fifth Discipline - Senge
>
> [Ref 12] Perceived Conversation Quality in Spontaneous Interactions, (https://arxiv.org/pdf/2207.05791)
>
> [Ref 13] Social Processes (https://arxiv.org/abs/2107.13576)
>
> [Ref 14] Capturing Interaction Quality in Long Duration (Simulated) Space Missions with Wearables (https://doi.org/10.1109/TAFFC.2022.3176967)

---

> ### Author Response · Authors · 2022-08-24
> **Fail-safes to guard against data abuse**
>
> Thank you for the comments, reviewer iFrJ! We are glad that you acknowledge that the steps we’ve taken to uphold the ethical best-practices set up a useful precedent for research of this nature.
>
> 1\.
> >  The more likely use of this sort of research is for private companies to sell tools that claim they can help build relationships… Will these companies likely limit themselves to the reasonable restraints the authors have developed, and how can this be assured?
>
> The scenario that the reviewer is describing would constitute a **legal violation** of our EULA. Overall, we have two types of fail-safes in place. We have already described in the documentation, and summarize below for convenience: I) the conditions of the EULA that establish grounds for a legal violation, II) manual checks before granting data access.
>
> ### Fail-Safe I: Establishing grounds for legal violation
> - Following point 6 in the consent form (Fig. 15, Datasheet), participants have only provided consent to share the data `with other researchers in the research community, only in the case of research that is substantially similar in purpose to the goal of this research project (analysis of community/network dynamics, analysis of social interaction in mingling scenarios) and only if these parties comply with the European Union General Data Protection Regulation (GDPR).`
>
> - Point 6 also states that `[t]he recorded data will not be made freely available to the general public… Any researchers requesting access to the data will be required to sign an End-User License Agreement (EULA) agreeing to keep the data private and to the responsible use of the data as described in point 6, as well as compliance with the GDPR. `
>
> - Under Section 9 of the EULA that the researchers sign for access ((https://doi.org/10.4121/20016194)[https://doi.org/10.4121/20016194], Appendix A), we clearly state:
> ```
> Any commercial use of the Dataset is strictly prohibited. Commercial use of the Dataset includes, but is not limited to: 
> - Proving the efficiency of commercial systems; 
> - Testing commercial systems; 
> - Using screenshots of subjects from the Dataset in advertisements; 
> - Selling data or making any commercial use of the Dataset;
> - Broadcasting data from the Dataset.
> Any violation of this clause will give rise to immediate legal prosecution by the Licensor.
> ```
> From the TUDelft Human Research Ethics Committee (HREC), we have confirmation that should a violation of these terms be detected, the university will take action along with the data protection agency. Note that claims by companies of the nature that “they can help build relationships” may go against GDPR principles like data minimisation, accuracy and transparency [Ref 1], and are at odds with our EULA.
>
> [Ref 1] [https://gdpr-info.eu/art-5-gdpr/](https://gdpr-info.eu/art-5-gdpr/)
>
> ### Fail-Safe II: Checks before granting data access
>
> To further reduce the likelihood of violations of our EULA, our process for granting access to the dataset is meticulous. Each request is manually reviewed by the team before granting access. In cases of doubt, for similar past datasets, we have looked to obtain more information about the requester’s organisation and intended use of the dataset. A decision is then made by a committee that has been formed for ConfLab in consultation with our TUDelft Data Steward. We also require the subject to have a permanent position at their organisation, to avoid situations where the data could be transferred to a non-compliant organisation. (Also see the last point of our response here: [https://openreview.net/forum?id=CNJQKM5cV2o&noteId=kZJSOLPQvq_](https://openreview.net/forum?id=CNJQKM5cV2o&noteId=kZJSOLPQvq_)).
>
> ---
>
> Note that we cannot assure complete compliance with the EULA beforehand, or that users will not make legal violations once they have access. However, we believe this would not be a valid criticism of our methodology; we have done what is possible to make such outcomes unlikely, or enable post-hoc action should they be detected.
> Nevertheless, we expect that companies will likely adhere to the terms in our EULA to avoid legal problems. We have also been having an open discourse with researchers at industrial research labs to ensure that researchers doing academic research in such labs can have access to the data for non-commercial purposes. This would be further monitored by companies’ internal ethics boards. Note that the copy-left license (EULA point 6) already requires any resulting insights from the use of the data to be shared back with the scientific community.
>
> Beyond all of this, we have also taken measures to reduce the likelihood of data abuse by directly incorporating ethical consideration in most aspects of our design, including the choice of camera view, and low-frequency audio recording.

---

### Author Response · Authors · 2022-08-16
**Common Clarification : Considerations behind the choice of camera perspective**

The reviewers have expressed some overlapping concerns related to the top-down video perspective in our dataset: its usefulness for keypoint detection (jq5N), its influence on annotation quality (PsHE), and the lack of emotional expressions, possibly facial (XnKf). The design of the video capture was an extended topic of discussion amongst the authors internally, and externally with the university Human Research Ethics Committee (HREC) and the General Chairs (GCs) of the conference the dataset was recorded at.  We provide common context for individual responses here:

### Purpose of the dataset

Contrary to PsHE’s opinion: `a superior approach would be to conduct either semi-natural recordings in a lab with superior ground truth`, datasets with semi-natural interactions in the lab provide no novelty (see discussion of the Panoptic Studio [24], also see L69-79).

To further clarify, the primary goal of this work is **not** to:
- improve upon such in-the-lab efforts or capture more semi natural interactions.
- solve keypoint detection from overhead views at the cost of violating the ecological validity of the interactions (real-world relationships within an international community.)
- @jq5N: rather, improving keypoint detection is a necessary consequence due to the aerial camera perspective that is needed for recording in these settings (see below).

Rather, the main purpose of ConfLab is to:
- enable the modelling and study of social interaction dynamics **in the natural ecology**, which in this case is an international community at a conference.
- address limitations surrounding lower fidelity in related datasets with respect to both quality of raw data, multimodal synchronization, and annotations (Table 1; we’re in fact the first to provide keypoint annotations).


### Ethical Considerations

We were advised by both the HREC and GCs against using the elevated side-view perspective to mitigate the capturing of faces. Faces constitute one the most sensitive personally-identifiable features. The idea is that inclusion of faces would make it harder to
- prevent misuse in downstream tasks for developing surveillance technologies with negative societal impact, e.g. the Duke MTMC dataset was retracted following an investigation by Financial Times (https://www.ft.com/content/cf19b956-60a2-11e9-b285-3acd5d43599e), which related to the presence of faces; it was also been linked to providing surveillance of Uighur Muslims in China (https://exposing.ai/duke_mtmc/).
- protect the participants who are part of a real research community; since the dataset doesn’t involve role playing or scripted conversations, the participants are being themselves.

### Technical Considerations

We had considered three potential camera perspectives: egocentric, elevated side-view, and top-down. Note that occlusion is unavoidable in any of these camera perspectives, being a natural characteristic of crowded settings. Our capture design goal was to find the best trade-off between minimising occlusion and interfering with the naturalness of the interaction (validity).
- The ego-centric perspective would be privacy invasive. Additionally such a setup is harder to scale with the number of participants since it requires an expensive camera per person (also see JhBB’s point about the cost).
- The elevated side view camera is in fact **worse** for occlusion issues than the top-down perspective in crowded scenes. This is because people’s bodies are blocked more easily in crowded scenarios from a side viewing angle, making it harder to capture gestures (see [15, Fig. 1, 2, 7]). Even people close to the cameras can be occluded if they are backing the camera.
- In contrast, the top-down perspective suffers the least from occlusions from other subjects in a crowded environment, allowing for the capture of gestures and lower extremities of the most number of people without intruding on the interaction. We’ve added examples comparing top- and side-views from ConfLab in the rebuttal revision (Fig. 6)

---

### Author Response · Authors · 2022-08-16
**New Revision and Supplementary Material**

Dear Reviewers,

Thank you all for your comments and suggestions! We are grateful for the feedback that has let us improve our manuscript. We have submitted a new revision of the main paper and appendices with the changes highlighted. We address the comments directly in our individual responses. Broadly, the main changes are as follows:

1.  Introduction rework: We clarifying the intrinsic trade-off between data fidelity, ecological validity, and ethical considerations for a data collection of this nature. We believe this might help address some comments from reviewers PsHE, jq5N, and XnKf.

2. Video for qualitatively visualizing the annotation quality: We believe that some comments about keypoint annotation quality from reviewer PsHE might simply stem from a misunderstanding. To provide context we have visualized a 2.5 min segment of keypoint annotations which is available in the supplementary material (`keypoints_annotations_sample.mp4`, which will also be added to the samples directory in the dataset on 4TU.ResearchData).

3. Adding information about the pilot studies we conducted for establishing inter-annotator agreement for our different annotations.

4. Clarifications / Improvements based on the 'minor points' feedback from reviewer PsHE and 'additional feedback' points from reviewer JhBB.

---

> ### Author Response · Authors · 2022-08-29
> **Updated supplementary revision with a report on intelligibility of low-frequency audio**
>
> Following a discussion with reviewer PsHE surrounding why we didn't use the audio for speaking status annotations ([https://openreview.net/forum?id=CNJQKM5cV2o&noteId=uwT47o84Yj-](https://openreview.net/forum?id=CNJQKM5cV2o&noteId=uwT47o84Yj-)), we've added a report on the intelligibility of the privacy-preserving audio to a revision of the supplementary material. We'd conducted this investigation for due diligence, since the point of recording at low frequencies was to preserve privacy.

---

### Meta-Review · Area_Chair_MUhH · 2022-09-03

**Recommendation:** Accept
**Confidence:** 4

**Metareview:**

The paper proposes a dataset with human interactions recorded in an ACM MM workshop. It involves multiple overhead, privacy-preserving videos, audio and wearable sensors. The author group has also provided continuous key points, speaking statuses and F-formation annotations. The dataset supports person/keypoint detection, speaking status detection, and F-formation detection. In a broader context, this dataset can be useful for many applications such as security surveillance and social signal processing.

The reviewers found the paper to be well written and consider the dataset to be somewhat useful to the scientific community. One of the reviewers is concerned about the size and quality of the dataset to be helpful in driving novel AI/ML approaches in the NeurIPS community.

The reviewers appreciate the authors' responses to their reviews and the changes implemented in the revised version of the manuscript. In particular, there has been a discussion related to potential ethical implications of this work. The reviewers seem to be satisfied with how the authors have addressed the ethical concerns. Hence, I recommend to accept this paper.

---

### Decision · Program_Chairs · 2022-09-16

Accept